# Development of FluoAHRL: A Novel Synthetic Fluorescent Compound That Activates AHR and Potentiates Anti-Inflammatory T Regulatory Cells

**DOI:** 10.3390/molecules29132988

**Published:** 2024-06-23

**Authors:** Natalija Jonić, Ivan Koprivica, Christos M. Chatzigiannis, Antonis D. Tsiailanis, Stavroula G. Kyrkou, Eleftherios Paraskevas Tzakos, Aleksandar Pavić, Mirjana Dimitrijević, Andjelina Jovanović, Milan B. Jovanović, Sérgio Marinho, Inês Castro-Almeida, Vesna Otašević, Pedro Moura-Alves, Andreas G. Tzakos, Ivana Stojanović

**Affiliations:** 1Department of Immunology, Institute for Biological Research “Siniša Stanković”—National Institute of the Republic of Serbia, University of Belgrade, 11108 Belgrade, Serbia; natalija.jonic@ibiss.bg.ac.rs (N.J.); ivan.koprivica@yahoo.com (I.K.); mirjana.dimitrijevic@ibiss.bg.ac.rs (M.D.); 2Section of Organic Chemistry & Biochemistry, Department of Chemistry, University of Ioannina, 45110 Ioannina, Greece; cmchatzigiannis@gmail.com (C.M.C.); antonis.tsiailanis@gmail.com (A.D.T.); stavroylakyrkoy@gmail.com (S.G.K.); 3Department of Biology, National and Kapodistrian University of Athens, 15772 Athens, Greece; etzakos@gmail.com; 4Laboratory for Microbial Molecular Genetics and Ecology, Institute for Molecular Genetics and Genetic Engineering, University of Belgrade, 11000 Belgrade, Serbia; pavicaleksandarr@gmail.com; 5Department of Otorhinolaryngology with Maxillofacial Surgery, Clinical Hospital Center “Zemun”, 11080 Belgrade, Serbia; andjelinakjosevski@yahoo.com (A.J.); majov@eunet.rs (M.B.J.); 6Faculty of Medicine, University of Belgrade, 11000 Belgrade, Serbia; 7Instituto de Biologia Molecular e Celular (IBMC), Universidade do Porto, 4200-135 Porto, Portugal; smarinho@i3s.up.pt (S.M.); ialmeida@i3s.up.pt (I.C.-A.); 8Instituto de Investigação e Inovação em Saúde (i3S), Universidade do Porto, 4200-135 Porto, Portugal; 9Department of Molecular Biology, Institute for Biological Research “Siniša Stanković”—National Institute of the Republic of Serbia, University of Belgrade, 11108 Belgrade, Serbia; vesna@ibiss.bg.ac.rs; 10Institute of Materials Science and Computing, University Research Center of Ioannina (URCI), 45110 Ioannina, Greece

**Keywords:** Aryl Hydrocarbon Receptor (AHR), T regulatory cell (Treg), inflammation, CYP1A1

## Abstract

Aryl Hydrocarbon Receptor (AHR) ligands, upon binding, induce distinct gene expression profiles orchestrated by the AHR, leading to a spectrum of pro- or anti-inflammatory effects. In this study, we designed, synthesized and evaluated three indole-containing potential AHR ligands (FluoAHRL: AGT-4, AGT-5 and AGT-6). All synthesized compounds were shown to emit fluorescence in the near-infrared. Their AHR agonist activity was first predicted using in silico docking studies, and then confirmed using AHR luciferase reporter cell lines. FluoAHRLs were tested in vitro using mouse peritoneal macrophages and T lymphocytes to assess their immunomodulatory properties. We then focused on AGT-5, as it illustrated the predominant anti-inflammatory effects. Notably, AGT-5 demonstrated the ability to foster anti-inflammatory regulatory T cells (Treg) while suppressing pro-inflammatory T helper (Th)17 cells in vitro. AGT-5 actively induced Treg differentiation from naïve CD4^+^ cells, and promoted Treg proliferation, cytotoxic T-lymphocyte-associated antigen 4 (CTLA-4) expression and interleukin-10 (IL-10) production. The increase in IL-10 correlated with an upregulation of Signal Transducer and Activator of Transcription 3 (STAT3) expression. Importantly, the Treg-inducing effect of AGT-5 was also observed in human tonsil cells in vitro. AGT-5 showed no toxicity when applied to zebrafish embryos and was therefore considered safe for animal studies. Following oral administration to C57BL/6 mice, AGT-5 significantly upregulated Treg while downregulating pro-inflammatory Th1 cells in the mesenteric lymph nodes. Due to its fluorescent properties, AGT-5 could be visualized both in vitro (during uptake by macrophages) and ex vivo (within the lamina propria of the small intestine). These findings make AGT-5 a promising candidate for further exploration in the treatment of inflammatory and autoimmune diseases.

## 1. Introduction

The Aryl Hydrocarbon Receptor (AHR) constitutes a highly conserved ligand-activated transcription factor [1,2]. AHR ligands typically contain aromatic rings, notably indoles, and their binding affinity to AHR, along with elicited responses, can exhibit considerable variation among tissues and cells across different species. These ligands may function as either agonists or antagonists of AHR, with sources encompassing exogenous elements such as pollutants and dietary compounds, as well as endogenous elements such as microbiota- and host-derived metabolites [1,2]. In its inactive state, AHR is sequestered in the cytoplasm as part of a multiprotein complex. Ligand binding releases AHR from this multiprotein complex facilitating its translocation to the nucleus. Once in the nucleus, AHR orchestrates the transcription of various genes, including phase I enzymes responsible for ligand metabolism and detoxification, such as CYP1A1 and CYP1B1 [1,2,3]. However, AHR possesses the capability to regulate the transcription of numerous genes associated with the immune response. This regulation occurs either through direct binding to specific DNA-binding sites in their regulatory regions or via interaction with diverse transcription factors [1,2]. For instance, pollutants like 2,3,7,8-tetrachlorodibenzo-p-dioxin (TCDD) can bind to AHR, influencing NF-κB signaling in dendritic cells [2]. AHR demonstrates the capacity to interact with both canonical and noncanonical NF-κB pathways, steering dendritic cells towards an anti-inflammatory phenotype [1,2,3,4].

Activation of AHR exerts a dual role in immune modulation, demonstrating both pro- and anti-inflammatory effects. This multifaceted impact is contingent upon various factors in a ligand-, cell-, tissue- and organism-dependent manner [1,3,5,6]. For instance, the tryptophan metabolism product 6-formylindolo[3,2-b]carbazole (FICZ) activates AHR and exerts pro-inflammatory action through the stimulation of T helper 17 (Th17) cells in delayed type hypersensitivity reaction and multiple sclerosis [5,6,7,8]. Conversely, certain compounds such as dietary compound indole-3-carbinol (I3C), the metabolite 2-(1′H-indole-3′-carbonyl)-thiazole-4-carboxylic acid methyl ester (ITE) and TCDD have demonstrated beneficial effects in treating inflammatory conditions, like inflammatory bowel disease, experimental autoimmune encephalomyelitis, dermatitis, psoriasis and type 1 diabetes [5]. Along these lines, compounds stimulating AHR have been identified to beneficially affect innate and adaptive immunity and prevent tumor development in a variety of cancer types, including glioma, making AHR an appealing prospective therapeutic target [9]. Presently, the mechanistic understanding of these AHR ligands predominantly relies on the activation of tolerogenic dendritic cells and T regulatory cells (Treg). Notably, kynurenine, a product of tryptophan metabolism, promotes both FoxP3^+^ Treg differentiation and tolerogenic dendritic cells [10,11]. AHR can also enhance the production of enzymes that convert vitamin A into retinoic acid, which in turn facilitates the differentiation of FoxP3^+^ Treg cells, especially in mucosal tissues. [12,13].

Additionally, AHR can promote the immunosuppressive function of Treg. It induces the expression of the ectoenzyme CD39, which works alongside CD73 to break down the pro-inflammatory extracellular ATP, thereby enhancing the suppressive function of Treg [14]. In addition to the overall stimulatory effect on Treg, AHR has specific roles in intestinal Treg. Herein, AHR is implicated in gut Treg development, as Treg-specific deletion of AHR leads to a marked decrease in Foxp3^+^ Treg in the gut, but not in other organs [15]. AHR is predominantly involved in the enhancement and maintenance of Treg suppressive functions while being redundant for their stability [16]. Additionally, colon epithelial cell AHR activation promotes Treg expansion in the lamina propria [17].

Recently, many efforts have been directed toward the synthesis of novel AHR ligands that can exert anti-inflammatory effects and that may be produced at a large scale [18]. Synthesis of small molecules characterized by dual functionality, fluorescence and binding affinity to the AHR remains an unexplored frontier with substantial potential in diverse applications such as biological imaging, diagnostics and therapeutics. To address this knowledge gap, this study endeavors to (i) develop small molecules with this dual modality; (ii) identify the compound demonstrating prominent anti-inflammatory properties among the synthesized compounds; and (iii) elucidate the specific mechanism(s) of the exerted anti-inflammatory effects. By developing a compound that combines these properties (immunomodulation and fluorescent emission), we can simplify the concept of a fluorescent probe–drug conjugate into a single, multifunctional molecule.

## 2. Results

### 2.1. Design and Synthesis of Novel Fluorescent AHR Ligands

In this study, our objective is to pioneer the development of novel AHR ligands characterized by a dual functionality: fluorescence in the near-infrared and binding to the AHR (Figure 1 and Figure 2; Appendix A). We have termed these compounds “FluoAHRL”. To achieve this innovative design, we adopted a systematic approach involving the generation of a donor-π-acceptor fluorescent scaffold. As a donor moiety for this, we incorporated well-established AHR binding scaffolds, thereby ensuring the integration of validated AHR-binding components into the novel FluoAHRL. This strategic synthesis aims to combine the advantageous properties of fluorescence, especially in the near-infrared region where light has high tissue depth of penetration and minimal absorption by intrinsic chromophores [19,20], with targeted AHR binding, opening new avenues for advanced molecular imaging and biological studies. As donors, we utilized the warheads resorcinol [21], indole-3-carbinol [22] and the N,N-dimethyl aniline ring contained in methyl yellow [23] (Figure 1A). Furthermore, in our design approach, we endeavored to develop rigid scaffolds that could also be compliant to operate as donor-π-acceptor fluorophores. The design was inspired by the binding of indoxyl-derived indirubin for which its bound structure to the AHR PAS-B domain has recently been unveiled, providing valuable insights for our design approach (pdbid: 7ZUB) [24]. As an acceptor, we harnessed the potential of the 2-(2-methyl-4H-chromen-4-ylidene) malononitrile (DCM) core. This choice was driven by the versatility of nitrile groups within the DCM structure, which are recognized as effective bioisosteres for carbonyl, hydroxyl, carboxyl and halogen functional groups [25]. Such bioisosteres are commonly encountered in ligand–protein interactions, adding a strategic dimension to our design approach. Moreover, the DCM can undergo condensation, facilitated by a Knoevenagel reaction, with diverse donors featuring aldehyde groups. We explored the incorporation of aldehydes such as 2,4-dihydroxybenzaldehyde, 3-formylindole and 4-(dimethylamino)benzaldehyde, mimicking the structural motifs of resorcinol, indole-3-carbinol/3-indoxyl and methyl yellow, respectively. This selection broadens the scope of our design by introducing varied functionalities through the aldehyde donors. Along these lines, we designed compounds AGT-4, AGT-5 and AGT-6 (Figure 1B). To verify the potential interaction of the designed compounds with the PAS-B of AHR we performed docking calculations using as a template the pdbid 7ZUB and indirubin (Figure 2; Appendix A). Interestingly, all designed compounds gave more favorable interaction calculation energies with respect to indirubin. Specifically, AGT-4 energy was −14,352 kcal/mol, AGT-5 was −13,219 kcal/mol and AGT-6 was −15,105 kcal/mol, whilst indirubin gave a binding score of −12,081 kcal/mol (Figure 2). The higher recorded affinity of the three compounds was due to the development of an array of interactions. AGT-4 formed hydrophobic and stacking interactions with His291 and Phe295 and also His337. Furthermore, Ser336 and Ser 365 developed favorable hydrogen bonding interactions with the nitrile groups of AGT-4 as well as hydrogen bonding of its phenol group with Gln 383. In AGT-5, Ser336 formed hydrogen bonding with nitrile from the 2-(2-methyl-4H-chromen-4-ylidene) malononitrile core. Also, the same core formed stacking interactions with Phe351 and with Tyr302 and Phe295. The indole ring of AGT-5 forms stacking interactions with His291 and Phe 324 and also hydrogen bonding interactions with Thr299. AGT-6 is stabilized in the bound state of AHR with an array of interactions. Specifically, the DCM core of AGT-6 formed stacking interactions with Phe295 and Phe351. Also, the nitrile group forms hydrogen bonds with Ser365, Ser346 and Ser336. Furthermore, the 4-(dimethylamino)benzyl group forms favorable stacking interactions with Phe324.

Having recorded the favorable interaction of the three ligands with AHR, we then calculated bioavailability parameters of the candidate compounds. Such parameters as i.e., absorption, distribution, metabolism and excretion, were investigated using the SwissADME web tool [26]. We evaluated gastrointestinal bioavailability, the blood–brain barrier permeation, P-glycoprotein-provided resistance and skin permeation and five different rule-based filters, with diverse ranges of properties inside of which the molecule is defined as drug-like. The findings, presented in Appendix A, demonstrate the drug-like properties of the compounds along with favorable calculated pharmacokinetic parameters. Having determined that the designed compounds exhibit a favorable interaction profile with AHR, along with desirable drug-likeness and calculated pharmacokinetics, we proceeded to synthesize them. The synthesis of the novel ligands was performed as depicted in Figure 1 and Appendix A. Briefly, the first step was to synthesize the DCM core, which was prepared by a three-step procedure, starting from 4′-hydroxyacetophenone. The synthesis of the AGT-4, AGT-5 and AGT-6 AHR ligands was successfully achieved through a Knoevenagel condensation of the DCM core with 2,4-dihydroxybenzaldehyde, 3-formylindole and 4-(dimethylamino)benzaldehyde, respectively, after heating to reflux overnight in acetonitrile and a catalytic amount of piperidine, followed by filtration in case the solid had come off, or otherwise purified by column chromatography. The structures of the synthesized compounds were characterized using ^1^H NMR, ^13^C NMR and LC-MS, and purity was evaluated by HPLC (see Appendix A).

Having the three compounds synthesized, we then aimed to validate their potential to operate as designed in the concept of the Donor-π-Acceptor and be fluorescent. For this, we recorded their photophysical properties. Interestingly, as it is illustrated in Figure 1C,D and Appendix A, all three compounds have emission in the near-infrared, and specifically, AGT-4 has emission at 842 nm, AGT-5 at 620 nm and AGT-6 at 610 nm. Considering the Donor-π-Acceptor structure and the recorded stoke shift values, the intramolecular charge transfer (ICT) can be suggested as the mechanism of fluorescence.

### 2.2. FluoAHRLs Potential to Modulate the Action of AHR

AGT-4, AGT5 and AGT-6 have the potential to interact with the PAS-B domain of AHR, based on the in silico studies and also bear optimal photophysical properties (Figure 1 and Figure 2: Appendix A). To evaluate the AHR modulatory capacity of the synthesized molecules, we took advantage of previously established AHR luciferase reporter cell lines [27,28,29,30,31]. In brief, THP-1 (human monocytes) and Caco-2 cells (human colon cancer epithelial cells) were exposed to the different molecules herein synthesized, or to a well-described AHR agonist, FICZ, as control [31]. As shown in Figure 3A–C, and in addition to FICZ, AGT-4 and AGT-6 significantly induced AHR activation both in THP-1 (after 4 h of incubation) and Caco-2 cells (after 4 h and 24 h), at the different concentrations tested. On the other hand, AGT-5 did not affect AHR transcriptional activity in THP-1 cells (Figure 3A) or Caco-2 cells at the 4 h treatment (Figure 3B) but activated AHR on Caco-2 cells after 24 h exposure (Figure 3C). Strikingly, AGT-5 could compete with FICZ-induced AHR activation after 4 h exposure in both THP-1 and Caco-2 cells (Appendix A), whereas, akin to AGT-4 and AGT-6, AGT-5 increased FICZ-induced AHR activation after 24 h exposure of Caco-2 cells (Appendix A). To further confirm AHR modulation, we performed gene expression analysis of an AHR-dependent gene, CYP1A [1,3] upon exposure of Caco-2 cells to 1.5 μM of the different molecules (concentration shown to elicit AHR activation in the luciferase reporter assays). Results obtained from qRT-PCR on Caco-2 cells show that all three compounds upregulated CYP1A1 mRNA expression after 24 h of exposure, AGT-6 being the most potent AHR agonist (similar to what was observed in the luciferase reporter assays), while AGT-5 showed the weakest agonistic ability (Figure 3D). Others have demonstrated that AHR modulation can occur indirectly by the presence and accumulation of endogenous AHR agonists, such as those originated via Tryptophan (Trp) metabolism [32,33,34]. To assess this possibility, we performed similar experiments, albeit by cultivating the cells in the presence of a Trp-free medium. In these conditions, as depicted in Appendix A, CYP1A1 mRNA upregulation was observed upon 24 h exposure of Caco-2 cells to the three tested compounds or FICZ.

### 2.3. Impact of FluoAHRLs on Immune Cell Polarization and Differentiation

To evaluate the impact of the synthesized compounds on the polarization and differentiation of immune cells, we first tested their activity on mouse peritoneal macrophages. Exposure of mouse peritoneal macrophages for 4 h to 1.5 µM of the different compounds led to increased Cyp1a1 mRNA (Figure 4A) and CYP1A1 protein (Figure 4B) expression levels, when compared to the DMSO control, confirming AHR activation. Similar results were obtained upon exposure to 1.5 µM of the AHR agonist control indoxyl-3-sulfate (I3S) [35] (Figure 4A,B). Taking advantage of the intrinsic AGT-5-emitted fluorescence (red), our results obtained by confocal microscopy confirmed induction of CYP1A1 expression in the AGT-5-treated peritoneal macrophages in vitro compared to DMSO-treated cells after 24 h of incubation (Figure 4C–E). Furthermore, we could observe an accumulation of AHR in the proximity of AGT-5, in contrast to a more diffuse and dispersed AHR expression in DMSO-treated cells (Figure 4F–K). Interestingly, as shown in Figure 4I–K, marked patterns of colocalization between the AHR and AGT-5 could be observed. Of note, to assess the potential cytotoxic effects of the newly synthesized compounds, mouse peritoneal macrophages were treated with increasing concentrations of AGT-4, AGT-5 and AGT-6 for 4 h and 24 h. Viability was not compromised in any tested concentration after 4 h of incubation (Appendix A). However, after 24 h of incubation, AGT-6 promoted cell death in the peritoneal macrophages (Appendix A). Therefore, for subsequent experiments, we focused on AGT-4 and AGT-5. To evaluate the impact of AGT-4 and AGT-5 on macrophage polarization, we exposed mouse peritoneal macrophages to the compounds and detected the expression of M1 and M2 markers (F4/80^+^CD40^+^ or F4/80^+^CD206^+^, respectively) by Flow Cytometry (FACS). The obtained results indicate that AGT-4 exposure increased the M1/M2 ratio, favoring M1 pro-inflammatory macrophages, whereas AGT-5 did not alter macrophage polarization in the testing conditions (Figure 4L). Of note, at the conditions used throughout the in vitro experiments, the fluorescence emission from AGT-4 and AGT-5 was not detectable by FACS, hence not interfering with the antibody stainings performed (Appendix A).

Similarly to the experiments performed with peritoneal macrophages, we evaluated the impact of AGT-4 and AGT-5 on CD4^+^ T cells isolated from mesenteric lymph nodes. Again, both ligands have shown no detectable fluorescent emission in designated channels for specific staining with antibodies (Appendix A). The isotype controls are depicted in Appendix A. During standard stimulatory conditions (in the presence of anti-CD3 and anti-CD28 antibodies—“complete” stimulation cocktail) [36]. AGT-5 preferably upregulated the Treg/Th17 ratio (Figure 5A). All in all, by comparing the effects of AHR ligands on macrophages and T cells, AGT-5 did not shift the M1/M2 ratio but stimulated the differentiation of Treg over Th17. What is more, the effect of AGT-5 on Treg was comparable to the AHR agonist control I3S [35] (Appendix A). These effects account for optimal anti-inflammatory capacity, and therefore AGT-5 was further examined in this study. To evaluate the possible effect of AGT-5 on the Treg/T helper 1 (Th1) ratio, purified CD4^+^ T lymphocytes from mesenteric lymph nodes were exposed to AGT-5 and the “complete” stimulation cocktail. The Treg proportion predominated in comparison to pro-inflammatory Th1 cells as observed by FACS (Figure 5B; Appendix A). AGT-5 also succeeded in stimulating in vitro differentiation of Treg from CD4^+^CD25^−^ naïve T cells in the absence of the anti-CD28 antibody (“incomplete” stimulation) (Figure 5C) and simultaneously increased their proliferation (Ki-67^+^ proportion) and production of interleukin-10 (IL-10) (Figure 5D). Also, when sorted Treg (CD4^+^CD25^high^) were exposed to AGT-5, their frequency increased, as well as the frequency of IL-10-producing Treg (Figure 5E and Appendix A). The stimulatory effect on Treg was dependent upon AHR activation, as the co-incubation with CH-223191, a known AHR antagonist [37], prevented the upregulation of the Treg proportion elicited by AGT-5 (Figure 5F and Appendix A). Also, the expression levels of Treg regulatory molecules were examined. The results revealed that AGT-5 upregulated the proportion of cells expressing cytotoxic T-lymphocyte-associated antigen 4 (CTLA-4), the inhibitory co-receptor for T cell activation (Figure 5G and Appendix A). These results suggest that AGT-5 can skew the differentiation of naïve CD4^+^ cells towards a regulatory phenotype, and to maintain their proliferation and anti-inflammatory activity by boosting IL-10 production and CTLA-4 expression.

As IL-10 production is regulated by signal transducer and activator of transcription 3 (STAT3) activation [38], we evaluated the level of total STAT3 and phosphorylated STAT3 (pSTAT3) in AGT-5- and DMSO-treated CD4^+^ cells. Results indicate that AGT-5 significantly increased the expression of total STAT3 (Figure 6), suggesting that this signaling molecule might be involved in the observed higher IL-10 expression (Figure 5D,E).

The phenomenon of AGT-5-mediated stimulation of Treg was also observed in human tonsillar cells, as AGT-5 efficiently and dose-dependently upregulated the proportion of Treg after 48 h of in vitro incubation (Figure 7A,B). Furthermore, AGT-5 also stimulated Treg proliferation in a dose-dependent manner, as determined by carboxyfluorescein succinimidyl ester (CFSE) staining (Figure 7A,C).

### 2.4. AGT-5 Safety Profile

Before engaging in the application of AGT-5 to mice, acute and chronic toxicity of AGT-5 was assayed on zebrafish embryos and compared to the known AHR ligand I3S. Lethal concentration 50 (LC50) values obtained in the toxicity assay, which indicate a dose that leads to the death of 50% of the embryos, revealed that the novel AHR ligand AGT-5 was completely safe (LC50 > 150 µM), in comparison to the commercially available I3S (LC50 = 11.5 µM). While AGT-5 did not elicit measurable toxic effects in zebrafish embryos during the five days of exposure, I3S induced lethality at doses higher than 25 µM (Figure 8A). At the dose of 5 µM (Figure 8B), I3S caused severe side effects in 4 out of 10 treated embryos (40%) such as life-threatening cardiotoxicity (pericardial edema and very slow heartbeat rate), nephrotoxicity, hepatotoxicity, skeletal deformations, while the rest of embryos (60%) had less serious malformations (less pronounced cardiotoxicity and/or hepatotoxicity) (Appendix A).

### 2.5. In Vivo Immunomodulatory Effects of AGT-5 in Healthy C57BL/6 Mice

To evaluate the impact of AGT-5 in vivo on T cells, AGT-5 was administered orally for five days to healthy C57BL/6 mice (Figure 9A). After oral application, AGT-5 fluorescence was found in the sections of the small intestine, where it localized below the epithelium, presumably within the lamina propria (Figure 9B,C). When evaluating the effect of AGT-5 on the immune response within mesenteric lymph nodes that drain the small intestine, it was found that AGT-5 significantly increased the proportion of Treg and decreased the proportion of Th1 cells, while there was no observable effect on Th17 cells (Figure 9D and Appendix A). Consequently, assessing the ratio of Treg vs. Th17 cells showed that AGT-5 exposure favored Treg (Figure 9E). Also, the proportion of Treg expressing Cyp1a1 in the mesenteric lymph nodes was increased after oral AGT-5 application (Figure 9E).

## 3. Discussion

This study unveils the development of three new molecules as AHR agonists (AGT-4, AGT-5 and AGT-6). Among those, AGT-5 displayed optimal anti-inflammatory activity, stimulating Treg differentiation and proliferation, and promoting their regulatory activity by increasing Treg IL-10 production and CTLA-4 expression in vitro. Furthermore, when administered orally to healthy C57BL/6 mice, AGT-5 increased the proportion of Treg and CYP1A1-expressing Treg in the mesenteric lymph nodes.

The expression of AHR differs across various cell types and tissues, and its ligand-binding affinities can also vary among species [1,2,39,40,41]. Therefore, in the current study, potential AHR ligands were tested by different assays in diverse cell types, including cells of mouse and human origin. In all the tested cell types, AGT-6 was the most potent AHR activator, assayed by AHR luciferase reporter assays and CYP1A1 gene and protein expression analysis, followed by AGT-4 and finally AGT-5. These results were supported by in silico docking studies. Also, in all cell types, short and long exposure to AGT-4 and AGT-6 showed AHR agonistic properties for both compounds. However, AGT-5 activity was time- and cell type-dependent. Longer exposure to AGT-5 was necessary for AHR activation in human Caco-2 cells, while mouse macrophages responded promptly to AGT-5, as observed by upregulated CYP1A1 mRNA and protein expression, similar to what was observed in the presence of AGT-4 or AGT-6. It has been documented that kinetics of AHR modulation and its effects differ between ligands, cells, tissues and organisms [42]. Not only was AGT-5 able to activate AHR in a species-independent manner, but it also exerted a similar immunomodulatory effect on mouse and human T cells, thereby opening up possibilities for its application in clinical trials.

The most studied function of AHR is to regulate xenobiotic-metabolizing enzymes such as CYP1A1 and CYP1B1. As shown by the luciferase reporter assays and gene expression analysis, all tested compounds—AGT-4, AGT-5 and AGT-6—show stimulatory effects on AHR activity. However, it has been described that some AHR ligands are toxic to cells, tissues and organisms. For example, TCDD (dioxin) is toxic at relatively low concentrations, as shown in in vitro and in vivo studies, causing life-threatening conditions such as toxic pancreatitis and hepatitis at concentrations of 20 μg/kg [43]. Similarly, I3S, the AHR ligand used in this study as an agonist in different assays, was lethal for zebrafish embryos at different concentrations, with a calculated LC50 of 11.5 μM. In contrast, AGT-5 was completely safe in the same model, causing no teratogenic or lethal effects when applied at concentrations that were 6-fold higher than those used for I3S. As for AGT-6, it reduced macrophage viability after 24 h of treatment and therefore this compound was omitted from further testing. Finally, AGT-4 was not cytotoxic for mouse macrophages, but, as it exerted pro-inflammatory properties, it did not fit into our search for immunosuppressive compounds. Taken together, our data indicate that AGT-5 might be a good candidate compound, as AHR activation elicited by AGT-5 was well-tolerated in vitro (by mouse macrophages) and in vivo (by mice), and non-toxic for zebrafish embryos. In addition, in silico evaluation by SwissADME web tool [26] indicated that AGT-5 possesses drug-like properties and a possibility of favorable pharmacokinetic parameters.

Both macrophages and T cells express AHR and their activity can be altered through AHR stimulation/inhibition [5,8,10,34]. Macrophages can be directed into an anti-inflammatory phenotype (downregulation of IL-6 and IL-1β) through AHR activation [44,45] AGT-4 preferentially promoted the M1 phenotype, while AGT-5 simultaneously stimulated both M1 and M2 phenotypes. Importantly, in AGT-5-treated macrophages, the M1/M2 ratio corresponded to the ratio observed in the non-treated cells. T cells can also respond to AHR ligands. Among T helper cells, Treg and Th17 cells have higher expression of AHR compared to Th1 and T helper 2 (Th2) cells [1,46] and are therefore more prone to AHR-mediated modulation. This higher AHR expression in Treg and Th17 is attributed to the action of TGF-β, a cytokine mandatory for their differentiation from naïve CD4^+^ cells [10]. Literature data shows that versatile AHR ligands specifically impact either Th17 or Treg [1,3,8]. According to our data, AGT-4 downregulates the proportion of Th17 with no impact on Treg. However, AGT-5 exposure acts differently on Treg and Th17 cells, favoring Treg and reducing Th17 proportions in vitro. One of the reasons for the observed upregulation of Treg could be the finding that AHR can upregulate FoxP3 transcription [47]. In addition, AGT-5 can stimulate the differentiation of naïve CD4^+^ cells into Treg, and it is also able to upregulate the number of already differentiated CD4^+^CD25^high^ Treg. This is consistent with the literature, where it was found that AHR activation by its high-affinity ligand TCDD expanded Treg in vivo [8]. Those Treg were fully functional and successfully suppressed the development of experimental autoimmune encephalomyelitis, experimental autoimmune uveoretinitis and spontaneous autoimmune diabetes [48,49]. Furthermore, I3C and ITE (both AHR agonists) were shown to attenuate colitis by upregulating Treg [46,50,51]. The endogenous AHR ligand kynurenine, which is produced during tryptophan degradation mediated by indoleamine 2,3-dioxygenase (IDO), is also able to promote Treg [10]. The observation that AGT-4, a structurally similar compound, did not upregulate the Treg proportion can be explained by the fact that agonist activity does not necessarily correlate with Treg promotion.1 For example, AHR agonist FICZ induces Th17 cells in vivo and therefore exacerbates encephalomyelitis in mice [8]. The outcomes of AHR activation in different immune cells are certainly ligand-driven and might stem from the capacity of AHR to interact with different transcriptional partners in different cellular contexts. Examples of those are found in the AHR interaction with transcription factors retinoic acid [52] and estrogen receptor [53], two receptors that alternatively influence Treg and Th17 cell differentiation [54,55].

Each T cell requires T cell receptor (TCR) stimulation in vitro to survive and proliferate, and in vitro, this is provided by anti-CD3 and anti-CD28 antibodies [56]. It is noteworthy that AGT-5 was able to promote Treg differentiation even in the absence of the co-stimulatory signal that comes from CD28 activation, suggesting that AGT-5 activates events that reconstitute CD28 signaling. What is more, AGT-5 not only aided Treg differentiation but also increased the proportion of Treg with suppressive functions. The immunosuppressive effects of Treg are mediated via the production of anti-inflammatory cytokines (IL-10 and interleukin-35), through cell-to-cell contact (CTLA-4, granzymes, perforins) or the depletion of ATP and generation of immunosuppressive AMP and adenosine [57]. Our results show increased proportions of IL-10^+^ Treg and CTLA-4^+^ Treg after AGT-5 treatment. This is in accordance with the literature data that demonstrate the ability of AHR to transactivate the IL-10 promotor in T cells and which was shown responsible for TCDD-mediated induction of Treg and Tr1 (regulatory cells that lack FoxP3 expression) [49]. Regulation of IL-10 expression by AHR is mediated by the Src-STAT3 signaling pathway [38]. As AGT-5 increased total levels of both pSTAT3 and STAT3, it can be postulated that STAT3 activation was involved in IL-10 upregulation observed in Treg. AGT-5 also targeted CTLA-4 expression in Treg. The suppressive function of CTLA-4 is exhibited by the inhibition of CD28, which represents a mandatory co-stimulatory signal for T cell activation [56]. Therefore, the upregulation of Treg that carry CTLA-4 by AGT-5 can be interpreted as stimulation of Treg inhibitory function. Although there is a direct link between AHR activation and CD39 expression [58], no effect of AGT-5 was observed on either CD39 or CD73 (enzymes involved in the generation of AMP and adenosine respectively) [14].

The AHR assumes a pivotal role in orchestrating the development of Treg within the Gut-Associated Lymphoid Tissue (GALT) [15]. This specialized population of Treg in the gastrointestinal tract is instrumental in upholding immune tolerance towards food and microbiota antigens [59]. The GALT, being susceptible to autoimmune responses, may witness initiation or perpetuation through mechanisms like molecular mimicry or bystander activation amid chronic gut inflammation [60]. Consequently, the strategic targeting and potential manipulating Treg in the GALT via AGT-5 present an effective avenue for fostering tolerance and mitigating autoimmune reactions. An additional noteworthy attribute of AGT-5 lies in its inherent fluorescence in the near-infrared, facilitating its detection in vivo. Therefore, FluoAHRL can provide visualization of AHR activation and distribution within living organisms, enable live-cell imaging studies and co-localization studies with other cellular components, and can facilitate high-throughput screening assays to identify novel compounds that modulate AHR activity. Small molecules that are both fluorescent and AHR-binders can be engineered into biosensors for diagnostic purposes. These biosensors could detect AHR activation levels, aiding in the diagnosis and monitoring of diseases associated with AHR dysregulation. This multifaceted approach not only contributes to the fundamental understanding of immune regulation in the gastrointestinal milieu but also holds translational potential for advancing precision immunotherapeutic strategies.

## 4. Materials and Methods

### 4.1. Procedures for the Synthesis of Potential AHR Ligands

Methylene chloride (CH_2_Cl_2_), ethyl acetate (EtOAc) and piperidine were purchased in anhydrous form and all commercially available chemicals were used without further purification, including these. Dry solvents were used for performing all reactions under anhydrous conditions unless otherwise specified. A syringe was used to transfer air- and moisture-sensitive liquids. Rotary evaporation was method of choice for concentration of organic solutions. Flash-column chromatography was performed with Acros Organics silica gel 60 (230–400 mesh), while thin layer chromatography (TLC) was performed with pre-coated Merck silica gel 60 F254 plates which were visualized by exposure to UV light and/or submersion in aqueous KMnO_4_/H_2_SO_4_. ^1^H/^13^C-NMR spectra were recorded either on a 500 MHz or 400 MHz Bruker Avance FT-NMR spectrometer, and 2D-NMR, HSQC and HMBC experiments were performed on a 500 MHz Bruker Avance FT-NMR spectrometer.

#### 4.1.1. General Procedure for the Synthesis of 2-Methyl-4H-chromen-4-one (**1**)

Synthesis of compound (**1**) was performed by modifying a recently reported protocol.62 Sodium (11.84 g, 511.56 mol) was added to a stirring solution of 2′-hydroxyacetophenone (8.8 mL, 73.08 mmol) in anhydrous ethyl acetate (50 mL) under nitrogen atmosphere and rose to reflux in a 100 mL double-necked round-bottomed flask with a reflux condenser. The mixture was stirred at room temperature (RT) for 14 h. After the reaction was complete (TLC control), ice-cold water was added and the pH value was set to 6 by using hydrochloric acid (HCl). This was followed by extraction, drying with anhydrous sodium sulfate, filtration and evaporation of the organic layer. The residue was dissolved in methanol (100 mL) and approximately 1 mL of HCl was added. The mixture was stirred at room temperature (RT) for 14 h. After the reaction was complete (TLC control), the solvent was removed and then extracted with water and dichloromethane, and the organic layer was dried with anhydrous sodium sulfate (Na_2_SO_4_) and filtered. The filtrate was evaporated to dryness and the residue was purified by column chromatography (30% ethyl acetate/hexane), at which time the pure compound (8.14 g, 51.16 mmol, 70%) was obtained as slightly orange crystals. ^1^H NMR (400 MHz, CDCl_3_) δ: 8.15 (dd, *J* = 1.6 Hz, 8 Hz, 1H, H-6), 7.60 (ddd, 1.6, 7.2, 8.6 Hz, 1Hz, H-8), 7.44–7.35 (m, 2H, H-7, H-9), 6.15 (s, 1H, H-2), 2.36 (s, 3H, H-12). ^13^C NMR (100 MHz, CDCl_3_) δ: 178.14 (C-3), 166.22 (C-1), 156.44 (C-10), 133.43 (C-8), 125.54 (C-6), 124.89 (C-7), 123.52 (C-5), 117.80 (C-9), 110.52 (C-2), 20.58 (C-12) (Appendix A).

#### 4.1.2. General Procedure for the Synthesis of 2-(2-Methyl-4H-chromen-4-ylidene)malononitrile (**2**)

Synthesis of compound (**2**) was performed based on a recently reported protocol [61]. To a stirred solution of (**1**) (537 mg, 3.307 mmol) in acetic anhydride (16 mL), malononitrile (262.36 mg, 3.969 mmol) was added and the reaction mixture was stirred vigorously under nitrogen atmosphere and refluxed for 14 h. The mixture was concentrated under a high vacuum to remove acetic anhydride. Water (20 mL) was then added and the mixture was refluxed for another 30 min. The mixture was extracted twice with CH_2_Cl_2_ (2 × 25 mL) and the organic layer was dried over Na_2_SO_4_, filtered and concentrated under reduced pressure to afford a black oil. The crude product was purified via flash column chromatography (Hexane-CH_2_Cl_2_ 1–3:1–2) to afford an orange-red solid (190 mg, 27.4%). ^1^H NMR (400 MHz, CDCl_3_) δ: 8.93 (dd, *J* = 1.6 Hz, 8 Hz, 1H, H-6), 7.73 (dd, 1.6, 7.2, 8.6 Hz, 1H, H-8), 7.44–7.35 (m, 2H, H-7, H-9), 6.73 (s, 1H, H-2), 2.44 (s, 3H, H-12). ^13^C NMR (100 MHz, CDCl_3_) δ: 161.89 (C-3), 153.37 (C-10), 152.98 (C-1), 134.93 (C-8), 126.11 (C-6), 125.35 (C-7), 118.93 (C-9), 117.72 (C-14), 116.85 (C-14), 115.63 (C-5), 105.53 (C-2), 62.37 (C-4), 20.66 (C-12) (Appendix A).

#### 4.1.3. General Procedure for Synthesis of AHR Ligands

The synthesis was performed by modifying a protocol from the literature [61]. A catalytic amount of piperidine was added to a solution of compound (**2**) (1 eq) and each aldehyde (1 eq) in anhydrous acetonitrile (2–10 mL) and the reaction mixture was heated to reflux under a nitrogen atmosphere for 14 h. After the reaction was completed, the solvent was evaporated under a high vacuum and the reaction mixture was purified by column chromatography, HPLC or in some cases precipitated as a solid precipitate and filtered directly by the reaction and washed off with cold diethyl ether.

#### 4.1.4. General Procedure for the Synthesis of AGT-4

Synthesis was performed between (**2**) (250 mg, 1.2 mmol) and 2,4-benzaldehyde dihydroxy (138.12 mg, 1.2 mmol) with the protocol described above. After filtration of the solution in a glass filter, the pure compound (AGT-4) settled as a dark red solid and was washed off with cold diethyl ether (149.1 mg, 34%). ^1^H NMR (400 MHz, DMSO-d_6_) δ 10.34 (s, 1H, 16-OH), 10.08 (s, 1H, 14-OH), 8.73 (dd, *J* = 8.3, 1.4 Hz, 1H, H-6), 7.97–7.85 (m, 2H, H12&8), 7.80 (dd, *J* = 8.5, 1.3 Hz, 1H, H-9), 7.65–7.55 (m, 2H, H-7&18), 7.19 (d, *J* = 16.0 Hz, 1H, H-11), 6.88 (s, 1H, H-2), 6.41 (d, *J* = 2.3 Hz, 1H, H-15), 6.34 (dd, *J* = 8.6, 2.3 Hz, 1H, H-17). ^13^C NMR (101 MHz, DMSO-d_6_) δ 162.04 (C-16), 160.24 (C-3), 159.27 (C-14), 153.28 (C-1), 152.53 (C-10), 135.62 (C-12), 135.48 (C-8), 130.28 (C-18), 126.48 (C-5), 125.04 (C-7), 119.50 (C-6), 118.15 (C-11), 117.67 (C-13), 116.79 (C-19), 114.76 (C-9), 114.16 (C-2), 108.83 (C-17), 105.59 (C-15), 58.43 (C-4). MS (HRMS): *m*/*z* for C_20_H_12_N_2_O_3_: [M]^−^ calculated 327.08, found 327.0776 (Appendix A).

#### 4.1.5. General Procedure for the Synthesis of AGT-5

The synthesis was performed between the compound (**2**) (250 mg, 1.2 mmol) and 3-indolecarbaldehyde (180 mg, 1.2 mmol) with the protocol described above. After the completion of the reaction, the solvent was removed and the compound was purified by column chromatography with CH_2_Cl_2_. Finally, the pure product (AGT-5) was received as a deep red solid compound cold (342 mg, 61%). ^1^H NMR (500 MHz, DMSO-d_6_): δ 11.91 (s, 1H, NH), 8.73 (d, *J* = 8.3 Hz, 1H, H-6), 8.23 (d, *J* = 7.7 Hz, 1H, H-19), 8.02 (d, *J* = 14.9 Hz, 2H, H-12&14), 7.89 (t, *J* = 7.8 Hz, 1H, H-7), 7.77 (d, *J* = 8.3 Hz, 1H, H-9), 7.57 (t, *J* = 7.7 Hz, 1H, H-8), 7.50 (d, *J* = 7.8 Hz, 1H, H-16), 7.29–7.19 (m, 3H, H-17,18&11), 7.09 (s, 1H, H-2). ^13^C NMR (126 MHz, DMSO-d_6_) δ 160.71 (C-1), 153.97 (C-10), 152.40 (C-15), 138.02 (C-20), 135.55 (C-12), 133.53 (C-14), 126.34 (C-8), 125.22 (C-6), 125.00 (C-17), 123.42 (C-11), 121.63 (C-19), 121.18 (C-9), 119.38 (C-5), 117.81 (C-18), 113.99 (C-16), 105.00 (C-2), 57.38 (C-4). MS (HRMS): *m*/*z* for C_22_H_13_N_3_O: calculated 335.11, found 336.37 [M + H]^+^ (Appendix A).

#### 4.1.6. General Procedure for the Synthesis of AGT-6

The synthesis was carried out between the compound (**2**) (250 mg, 1.2 mmol) and the 4-dimethylamino cinnamaldehyde (210.3 mg, 1.2 mmol) with the protocol described above. After evaporation under reduced pressure, the solid residue was dissolved in acetonitrile and purified with column chromatography (from 100:0 CH_2_Cl_2_:MeOH to 98:2 CH_2_Cl_2_:MeOH) where the desired product (AGT-6) was received as a dark purple solid (151.7 mg, 38.5%). ^1^H NMR (400 MHz, DMSO-d_6_) δ 8.71 (d, *J* = 8.4 Hz, 1H, H-6), 7.90 (t, *J* = 7.7 Hz, 1H, H-8), 7.76 (t, *J* = 11.5 Hz, 1H, H-7), 7.65–7.57 (m, 2H, H16&20), 7.46 (d, *J* = 8.2 Hz, 1H, H-9), 7.11–7.04 (m, 1H, H-14), 6.94 (dd, *J* = 27.4, 12.7 Hz, 2H, H-13), 6.84 (s, 1H, H-2), 6.80–6.63 (m, 3H, H-17&19, H-12), 3.04–2.94 (s, 6H, H-21). ^13^C NMR (101 MHz, CDCl_3_) δ 158.34 (C-1), 152.76 (C-10), 152.63 (C-3), 152.40 (C-18), 151.88 (C-14), 141.93 (C-12), 140.91 (C-8), 139.86 (C-11), 134.29 (C-16), 134.18 (C-20), 129.96 (C-15), 129.20 (C-7), 129.08 (C-6), 128.52 (C-5), 128.06 (C-8), 125.73 (C-13), 117.50 (C-22), 117.35 (C-22), 116.43 (C-9), 116.22 (C-17), 112.96 (C-18), 112.50 (C-2), 112.08, 105.75, 105.30, 77.38 (C-2), 40.50 (C21), 40.19 (C-21). MS (HRMS): *m*/*z* for C_24_H_19_N_3_O calculated 365.15, found 366.19 [M]^+^ (Appendix A).

### 4.2. HPLC Analysis

The purity of the compounds was evaluated by an Agilent analytical HPLC chromatography using a column: InfinityLab Poroshell 120 EC-C18, 4.6 × 150 mm and a diode array detector (DAD). The specific resolution had been observed at 254 nm. A gradient solvent system (A:5%, B:95% to A:100%, B:0% into 7 min and then A:100%, B:0% to A:5%, B:95% into 3 min), made up of A: ACN + 0.1% formic acid and B: H_2_O + 0.1% formic acid was used with a steady flow of 1.0 mL/min. A solvent ratio of 0.5% DMSO/Methanol was used as a blank. The HPLC analysis conducted for the AGT-4 compound revealed the targeted peak appearing at a retention time of 7.155, with a determination percentage of 97% (Appendix A). Similarly, for the compound AGT-5, the elution occurred at retention time 8.309, with a percent determination of 100% (Appendix A). Lastly, the analytical HPLC of the compound AGT-6 showed elution at retention time 9.346, with a percent determination of 95% (Appendix A). Therefore, all three synthesized compounds are >95% pure by HPLC analysis. Overlaid traces for the blank and all three FluoAHRLs are shown in Appendix A (the detector was set at 500 nm).

### 4.3. Protein Structure Preparation

Protein Data Bank (PDB) was used a source of cryo-EM structure of the indirubin-bound Hsp90-XAP2-AHR complex (PDB ID 7ZUB). Optimization of the structure was performed using the Protein Preparation Wizard in Maestro (Schrödinger. New York: Schrödinger Release, 2017–2021). Several steps were implemented during this procedure: adding missing hydrogens, filling in missing side chains and loops (Prime). In order to check for the protonation state of ionizable protein groups (pH 7.2) OPLS_2005 force field PROPKA was used [62]. Finally, the value of convergence of the RMSD of 0.3 Å was used to minimize the system.

### 4.4. Ligand Structure Preparation

All ligands were prepared using LigPrep (LigPrep, version 3.4; Schrödinger, LLC.: New York, NY, USA, 2017) available in Schrödinger Suites. The force field adopted was OPLS_2005 and Epik 3.9 (Schrödinger, 2017-1) was selected as an ionization tool at pH 7.2 ± 2.0 and the maximum number of conformers generated was set at 32.

### 4.5. Induced-Fit Docking

Molecular docking studies were performed by utilizing the Induced Fit Docking (IFD) method (Induced Fit Docking, Schrödinger Software Release 2017-1. Schrodinger Press; New York, NY, USA: 2017). The grid boxes for the binding sites of Chain D were built considering the co-crystallized ligand analog (3~{Z})-3-(3-oxidanylidene-1~{H}-indol-2-ylidene)-1~{H}-indol-2-one as a centroid.

The van der Waals scaling factor was adjusted to 0.5 in the initial stage of docking for both the receptor and the ligand. Side chains of residues within 5 Å of the ligand were optimized during the Prime refinement step. Each ligand docking generated a maximum of 20 poses, which were subsequently subjected to redocking using XP mode (docking poses are shown in Appendix A).

### 4.6. UV-Vis and Fluorescence Spectroscopy

UV-Vis and fluorescence spectroscopy were conducted using an Edinburg DS5 spectrophotometer and an Edinburg FS5 fluorometer, respectively. UV-Vis spectra were recorded in a 1 cm quartz cell at 37 °C, with a constant probe concentration of 10 μM for all measurements. Similarly, fluorescence spectra were recorded at the same temperature and slit settings to 5, also maintaining a consistent probe concentration of 10 μM in DMSO as a solvent throughout the experiments.

### 4.7. Cell Isolation and Treatment

Cells from the mesenteric lymph node were obtained by passing the tissue through a cell strainer (40 μm). Cell suspensions were centrifuged at 550× *g* for 5 min and obtained pelleted cells were finally resuspended in phosphate-buffered saline (PBS) containing 3% fetal calf serum (FCS). They were then incubated with a biotin-conjugated anti-mouse CD4 antibody (diluted 1:60, eBioscience, San Diego, CA, USA). After being washed with PBS, the cells were placed in PBS with 0.5% BSA and 2 mM EDTA which contained BD IMagTM Streptavidin Particles Plus–DM (1:20, BD Biosciences, Bedford, MA, USA). CD4^+^ cells were purified through attachment of the beads to the magnet; the process was repeated 3 times (8 min incubations). Cells were resuspended in RPMI 1640 medium containing 10% FCS, 2 mM L-glutamine, 25 mM HEPES, 0.02 mM Na-pyruvate and 5 μM β-mercaptoethanol, 1% penicillin and streptomycin (all from PAA Laboratories, Pasching, Austria). “Complete” stimulation cocktail (plate-bound anti-mouse CD3 (1 μg/mL) and soluble anti-mouse CD28 (1 μg/mL) antibodies) was used to stimulate the cells (Thermo Fisher Scientific, Waltham, MA, USA) and they were treated with AHR ligands for 48 h. CD4^+^CD25^−^ (naïve T cells) and CD4^+^CD25^high^ were purified by sorting with FACS Aria III (BD Biosciences, Bedford, MA, USA); details can be found further down in the flow cytometry paragraph. CD4^+^CD25^−^ cells were stimulated either with the “complete” or “incomplete” (only the anti-mouse CD3 antibody) stimulation cocktail and AGT-5 for 48 h in the presence or absence of AHR inhibitor CH-223191 (Sigma-Aldrich, St. Louis, MO, USA). CD4^+^CD25^high^ cells were kept in a medium with or without AGT-5 for 48 h and then stimulated with Cell Stimulation Cocktail (plus protein transport inhibitors) (eBioscience) for 4 h before the staining.

Peritoneal cells were collected after injection of cold PBS into the peritoneal cavity of mice. After centrifugation at 550× *g* for 5 min cells were placed in RPMI + 5% FCS and treated with AHR ligands.

Human tonsillar tissue samples were acquired during tonsillectomy at the Clinical Hospital Center “Zemun”, Belgrade, Serbia, from a 21-year-old female patient. This procedure was performed with the approval of the Ethics Committee of Clinical Hospital Center “Zemun”, Belgrade, Serbia (App. No 14/1, date 27 September 2022). Cells were obtained by passing the tissue through a cell strainer (40 μm), after which the cell suspension was centrifuged and the obtained cells were finally resuspended in RPMI containing 5% FCS. The cells were then incubated with AGT-5 for 48 h at 37 °C.

### 4.8. Viability Assay

Peritoneal cells were allowed to adhere to a 96-well plate (25 × 10^4^ cells/well) for 2 h, after which the culture medium was removed. The cells were then incubated with growing concentrations of AGT-5, AGT-4, AGT-6 (0.19–1.5 µM) or DMSO (0.0006–0.005% *v*/*v*) dissolved in RPMI containing 5% FCS for the duration of 4 h or 24 h at 37 °C. The medium was then removed, and 0.5 mg/mL 3-(4,5-dimethylthiazol-2-yl)-2,5-diphenyltetrazolium bromide (MTT, Sigma-Aldrich) was added. The plates were incubated at 37 °C for 30 min, after which the MTT solution was removed and DMSO was added. Absorbance was then measured at 540/674 nm using a Biotek Synergy H1 Microplate Reader (Agilent, Santa Clara, CA, USA).

### 4.9. Luciferase Assays

Luciferase activity measurement was performed as a readout for AHR activation, as previously described [3,4,5,6,7]. Briefly, THP-1 cells differentiated into macrophages, and Caco-2 AHR reporter cells [28,30] were exposed to the different compounds during different exposure periods. Subsequently, cell lysates were collected by harvest in reporter lysis buffer (Promega, Madison, WI, USA). The quantification of luciferase activity in the resulting supernatants was carried out using the Luciferase Assay System (Promega) in accordance with the manufacturer’s instructions. The luciferase activity data were adjusted to the protein amount determined for each sample, following the protocol of the Pierce BCA Protein Assay Kit (Thermo Fisher Scientific). The results are presented as fold induction upon normalization to the solvent control DMSO (AppliChem GmbH, Darmstadt, Germany).

### 4.10. Flow Cytometry

Cell surface molecules were detected on viable cells dispersed in Flow Cytometry Staining Buffer (eBioscience), using the antibodies displayed in Appendix A. Cell Stimulation Cocktail (plus protein transport inhibitors) (eBioscience) was used to treat the cells for the subsequent intracellular cytokine staining. After 4 h at 37 °C the cells were fixed in 2% paraformaldehyde (PFA) for 15 min at RT. Permeabilization of cells was performed using Permeabilization buffer (Thermo Fisher Scientific). After 30 min of permeabilization, cells were incubated with the antibodies displayed in Appendix A. Donkey anti-rabbit Alexa Fluor™ 488 (Abcam, Cambridge, UK) was kindly provided by Dr. Irena Lavrnja, Institute for Biological Research “Siniša Stanković”.

Regulatory T cells (Treg) were detected by Mouse Regulatory T cell Staining Kit according to the manufacturer’s instructions (eBioscience). The same protocol was applied for Ki-67 detection. Each staining was performed for 40 min at 4 °C. Isotype-matched controls were included in all experiments (eBioscience) (Appendix A). Cells were analyzed on FACS Aria III using either BD FACSDiva software version 8.0 or FlowJo v.10.10.0 software.

For the detection of cell proliferation, tonsillar cells were stained with CFSE (1 μM) (Thermo Fisher Scientific) for 15 min at 37 °C. Cells were washed twice in PBS and then placed in the U-bottom 96-well plate.

Cell sorting was also performed on FACS Aria III using anti-CD4 FITC antibody alone or anti-CD4 eF450 and anti-CD25 Alexa Fluor™ 488 antibodies to select for CD4^+^CD25^−^ and CD4^+^CD25^high^ cells.

### 4.11. Immunocytochemical Analysis

Peritoneal cells (8 × 10^4^ cells/well) were seeded on four-well chamber slides (Thermo Fisher Scientific) for 2 h, after which the slides were washed, leaving the peritoneal macrophages adhered to the plastic. The cells were then incubated in RPMI containing 5% FCS with AGT-5 (0.75 µM) for 24 h at 37 °C. At the end of the treatment, cells were washed three times in PBS and fixed in 4% PFA for 20 min. After washing, the cells were mounted with Fluoromount-G™ mounting medium, and as AGT-5 is a FluoAHRL with an excitation maximum of 540 nm and an emission maximum of 620 nm, its localization within the cells was documented using a confocal SP5 Leica microscope (Leica Microsystems GmbH, Wetzlar, Germany).

Conversely, for further immunocytochemical analysis, after being fixed and washed, the cells were permeabilized with 0.5% TritonX-100 in PBS for 20 min. Nonspecific binding of antibodies was reduced through the incubation step in PBS containing 5% BSA for 1h.

For immunocytochemical detection of AHR, incubation with primary mouse anti-AHR antibody (diluted 1:20 in PBS containing 1% BSA, Thermo Fisher Scientific) was performed overnight at 4 °C. After thorough washing with PBS, cells were incubated with Alexa Fluor 488 secondary donkey-anti-mouse antibody (Thermo Fisher Scientific) (diluted 1:200 in PBS containing 1% BSA) for 30 min at RT. After washing, cells were mounted with Fluoromount-G™ mounting medium, and the expression of AHR within the cells was documented using a confocal SP5 Leica microscope (Leica Microsystems GmbH).

For immunocytochemical detection of CYP1A1, incubation with primary rabbit anti-CYP1A1 antibody (diluted 1:100 in PBS containing 1% BSA, Thermo Fisher Scientific) was performed overnight at 4 °C. Cells were washed three times and then incubated with secondary goat-anti-rabbit antibody Alexa Fluor 488 (1:200 in PBS + 1% BSA) for 30 min at RT. Fluoromount-G™ mounting medium was used to finalize the staining and prepare the slides for analysis on a DM4B fluorescent microscope (Leica Microsystems GmbH). CYP1A1 signal intensity was quantified using Leica LAS AF intensity tool (Leica Microsystems GmbH). 80 cells per group were included in the regions of interest (ROIs) and intensity values were presented as means ± SEM.

### 4.12. qRT-PCR

A quantitative reverse transcriptase–polymerase chain reaction (qRT-PCR) assay was performed to assess changes in mRNA expression. For Caco-2 cells, after being exposed to different compounds for various time points, total RNA was extracted following the manufacturer’s instructions of the RNASpin Mini kit (Cytiva). RNA quality and concentration were accessed by spectrometry in NanoDrop One (Thermo Fisher Scientific). The synthesis of the complementary DNA (cDNA) was performed using the qScript cDNA Synthesis Kit (Quantabio, Beverly, MA, USA), according to the manufacturer’s protocol, on an XT96 thermocycler (VWR). The qRT-PCR protocol was executed using the enzymatic action of iTaq Universal SYBR (Bio-Rad, Hercules, CA, USA), and intercalant fluorescence was quantified in the CFX384 equipment (Bio-Rad) as follows: 3 min at 95 °C followed by 40 cycles of 10 s at 95 °C and 30 s at 60 °C. The average threshold of the triplicate reactions was used for all subsequent calculations using the DDC method [63,64]. The expression results for each sample were corrected for glyceraldehyde-3-phosphate dehydrogenase (GAPDH) or β-actin mRNA expression. The qRT-PCR data was generated from independent experiments, each with three biological replicates. The primers were acquired from Thermo Fisher Scientific, and their nucleic sequences are listed in Appendix A. For mouse peritoneal macrophages exposed to different compounds for 4 h, total RNA was extracted following the cell disruption in TriReagent (Metabion, Martinsried, Germany). After centrifugation with chloroform at 12,000× *g*, total RNA was isolated from the aqueous layer and reverse transcribed using random hexamer primers and MMLV reverse transcriptase, according to the manufacturer’s instructions (Fermentas, Vilnius, Lithuania). PCR amplification of cDNA was carried out in a Real-time PCR machine (Applied Biosystems, Woolston, UK) using SYBRGreen PCR master mix (Applied Biosystems) as follows: 10 min at 50 °C, 10 min at 95 °C, followed by 40 cycles of 15 s at 95 °C and 60 s at 60 °C. Accumulation of PCR products was detected in real-time and the results were analyzed with 7500 System Software (Thermo Fisher Scientific).

### 4.13. Western Blot

Peritoneal cells were allowed to adhere to a 24-well plate (2 × 10^6^ cells/well) for 2 h, after which the culture medium was removed. The cells were then incubated with AGT-4, AGT-5, AGT-6, I3S (1.5 µM) or DMSO (0.005% *v*/*v*) dissolved in RPMI containing 5% FCS. After 4 h of incubation, the medium was removed and the samples were dissolved directly in a western blot lysis buffer (containing 62.5 mM Tris–HCl (pH 6.8), 2% SDS, 50 mM DTT, 10% glycerol, with the Protease Inhibitor Cocktail (all from Sigma-Aldrich)).

Purified CD4^+^ cells were stimulated with a “complete” stimulation cocktail, IL-2 (10 ng/mL) and TGF-β (2 ng/mL) (both from R&D Systems, Minneapolis, MN, USA), and cultured with AGT-5 (1.5 µM) or DMSO (0.0025% *v*/*v*) for 24 h. The cells were then placed in TriReagent (2.5 × 10^6^ cells/sample). After the addition of chloroform and centrifugation at 12,000× *g*, proteins were isolated from the protein fraction following the manufacturer’s instructions and dissolved in the western blot lysis buffer.

In order to detect protein of interest using western blot, electrophoresis on 12% SDS–polyacrylamide gel was first performed. The protein samples were electro-transferred from the gel onto polyvinylidene difluoride membranes by using a semi-dry blotting system (Semi-Dry Transfer Unit, GE Healthcare, Buckinghamshire, UK). The membrane was blocked with PBS + 0.1% Tween-20 + 5% BSA. Membranes were exposed to specific antibodies diluted in PBST with 1% BSA. HRP conjugated anti-rabbit IgG (1:5000, Invitrogen) was used as a secondary antibody for anti-mouse STAT3 (1:750, Cell Signaling Technology), anti-mouse phospho-STAT3 (1:1000, Invitrogen) and anti-mouse CYP1A1 (1:750, Invitrogen). For β-actin detection, a HRP conjugated anti-mouse IgG (1:4000, Invitrogen) was used (1:1000, Abcam). Membranes were exposed to Immobilon Western Chemiluminescent HRP Substrate (Millipore, Billerica, MA, USA). iBright™ FL1500 Imaging System (Invitrogen) was used to detect the signal. Fiji software (https://imagej.net/software/fiji/downloads, accessed on 1 May 2024) was used to perform densitometry [65]. The quantification of targeted proteins were calculated relative to the presence of β-actin or the quantity of their non-phosphorylated protein forms.

### 4.14. Toxicity Assessment in the Zebrafish (Danio rerio) Model

All experiments involving zebrafish were performed in compliance with the European directive 2010/63/EU and the ethical guidelines of the Guide for Care and Use of Laboratory Animals of the Institute of Molecular Genetics and Genetic Engineering, University of Belgrade. Wild type (AB) zebrafish embryos were kindly provided by Dr. Ana Cvejić (Wellcome Trust Sanger Institute, Cambridge, UK) and maintained in zebrafish facility at standard 14:10-h light-dark photoperiod and at 28 °C. Commercial dry food (SDS300 granular food; Special Diet Services, Essex, UK and TetraMinTM flakes; Tetra, Melle, Germany) was used for twice a day feeding in addition to *Artemia nauplii* zooplankton which was added daily. Stock solutions of test substances used in the zebrafish assays were made in DMSO. Toxicity evaluation of AGT-5 as a new AHR ligand and I3S as an already known AHR ligand in the zebrafish model was carried out following the general rules of the OECD Guidelines for the Testing of Chemicals (OECD, 2013, Test No. 236) and the published protocols [66]. Briefly, wild type zebrafish embryos were distributed into 24-well plates (10 embryos per well) containing 1 mL E3 medium (5 mM NaCl, 0.17 mM KCl, 0.33 mM CaCl_2_ and 0.33 mM MgSO_4_ in distilled water). Embryos at 6 h post fertilization (hpf) stage were treated with five different concentrations of the tested compounds (5, 25, 50, 100 and 150 µM), and inspected for 22 toxicological endpoints (listed in Appendix A) under a Zeiss Stemi 508 Stereomicroscope (Carl Zeiss Meditec AG, Oberkochen, Germany), every day until 120 hpf. DMSO (0.25%) was used as negative control. Dead embryos were recorded and discarded every 24 h. Experiments were repeated twice. At 120 hpf, 0.1% (*w*/*v*) tricaine solution (Sigma-Aldrich) was added to the wells to anesthetize embryos. They are then photographed and froze at −20 °C for 24 h. Using the ToxRat program, the LC50 values (the dose causing the mortality of 50% of embryos) were determined.

### 4.15. Mice and AGT-5 Treatment

C57BL/6 mice were bred and maintained at the Animal Facility at the Institute for Biological Research “Siniša Stanković”, National Institute of the Republic of Serbia, University of Belgrade, with free access to food and water, and hiding structures added for environmental enrichment. All experiments were approved by the Veterinary Administration, Ministry of Agriculture, Forestry and Water Management, Republic of Serbia (App. No 119-01-4/11/2020-09), and were in accordance with the Directive 2010/63/EU on the protection of animals used for scientific purposes. AGT-5 was dissolved in DMSO to a stock solution of 50 mg/mL and then dissolved in sesame oil and applied at 10 mg/kg bw orally (approximately 0.25 mg per animal) for five days. Ex vivo analysis of the mesenteric lymph nodes and the small intestine was performed one day after the final AGT-5 administration.

### 4.16. Histological Analysis

Once the small intestine was harvested, approximately 1 cm in length was cut off and further used for histological analysis. The tissue was initially kept in 4% buffered formaldehyde until it was subjected to a set of different ethanol dilutions in the following order: 30%, 50%, 70% (all 2 × 30 min), 96% and 100% (2 × 1 h). This was followed by submerging the tissue in xylol for 2 × 3 min, after which it was placed into the first batch of paraffin (for 1 h) and then transferred into the second paraffin overnight. The next day it was embedded in the final paraffin and left at RT to consolidate. Using a microtome, the tissue was sectioned and the samples were analyzed with confocal microscopy.

### 4.17. Statistical Analysis

Data are presented as mean ± SD. The significance of differences between groups was determined by a two-tailed Student’s *t*-test. Differences are regarded as statistically significant if *p* < 0.05 (*), *p* < 0.01 (**) or *p* < 0.001 (***). Statistical analyses were performed using GraphPad Prism 5 software (GraphPad Software, Inc., La Jolla, CA, USA).

## Figures and Tables

**Figure 1 molecules-29-02988-f001:**
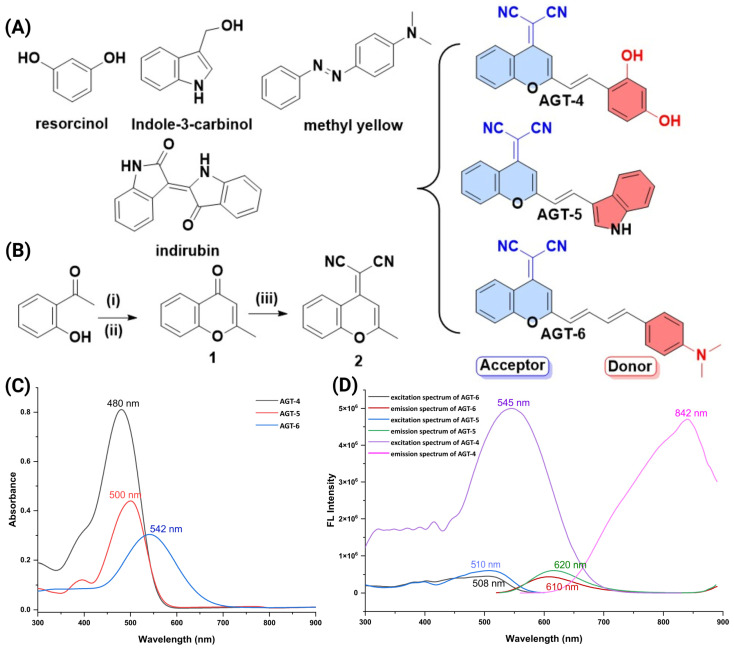
(**A**) Known AHR ligands. (**B**) Synthesis of the three novel fluorescent AHR ligands AGT-4, AGT-5 and AGT-6, based on resorcinol, indole-3-carbinol and methyl yellow scaffolds. Reagents and solvents: (i) NaH, EtOAc, THF, −5 °C, 15 min; (ii) Conc. HCl, MeOH, RT, 15 h; (iii) malononitrile, Ac_2_O, reflux, 15 h; in condensation reactions, the conditions are: corresponding benzaldehyde, piperidine, MeCN, reflux, 12 h. (**C**) Absorption spectra and (**D**) fluorescence spectra of compounds AGT-4, AGT-5, AGT-6 (10 µM), in dimethyl sulfoxide (DMSO), at 37 °C (step 2, ExBw: 5, EmBw: 5).

**Figure 2 molecules-29-02988-f002:**
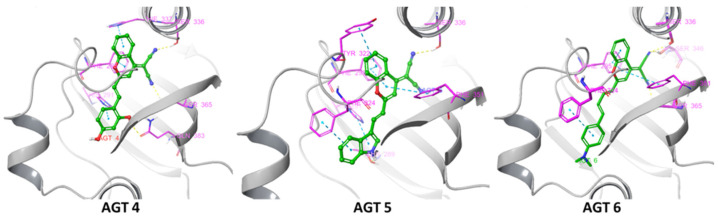
Docking poses of AGT-4, AGT-5 and AGT-6 in the PAS-B domain of AHR (pdbid: 7ZUB). The ligands are colored green, the interacting amino acids within 5Å are depicted in purple, and the protein is illustrated with grey ribbons.

**Figure 3 molecules-29-02988-f003:**
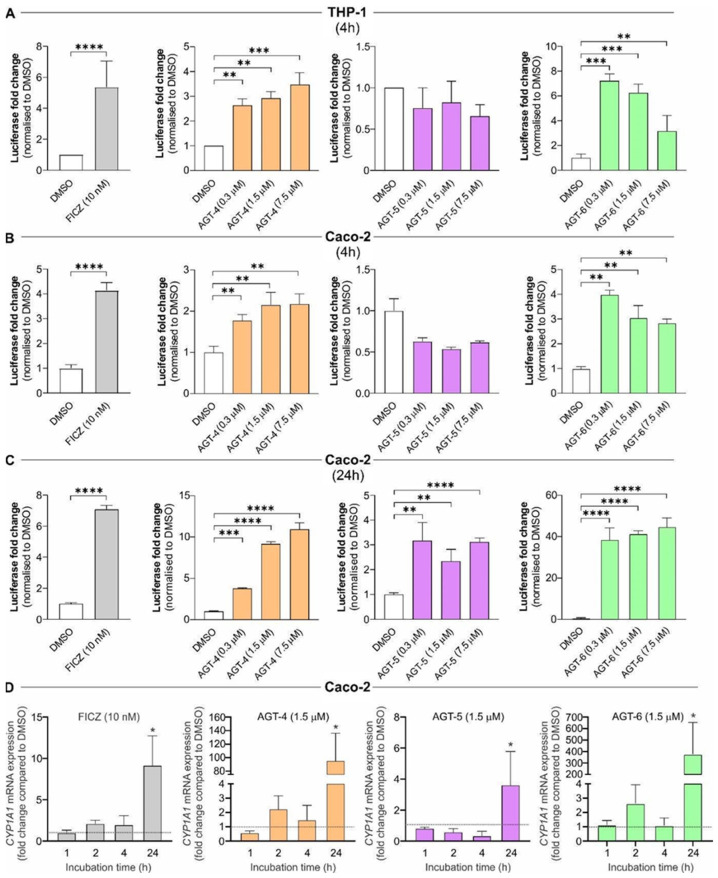
AHR modulation by newly synthesized molecules FluoAHRLs (AGT-5, AGT-4 and AGT-6). AHR luciferase reporter cell lines and AHR-dependent gene (CYP1A1) expression analysis were used to evaluate AHR activation after the treatment with the putative FluoAHRL, AGT-5, AGT-4 and AGT-6. FICZ was used as a positive control. Measurement of luciferase activity served as a readout for AHR activation and values were normalized to the results obtained from DMSO-treated cells. (**A**) Luciferase activity in THP-1 cells after 4 h of treatment. (**B**) Luciferase activity in Caco-2 cells after 4 h of treatment. (**C**) Luciferase activity in Caco-2 cells after 24 h of treatment. (**D**) CYP1A1 mRNA expression in Caco-2 cells was measured by qRT-PCR after 1 h, 2 h, 4 h or 24 h of treatment, normalized to the expression of GAPDH and then normalized to the values obtained in DMSO-treated cells at indicated time points. * *p* < 0.05, ** *p* < 0.01, *** *p* < 0.001, **** *p* < 0.0001 was considered as a statistically significant difference between AHR ligand-treated cells and DMSO-treated cells.

**Figure 4 molecules-29-02988-f004:**
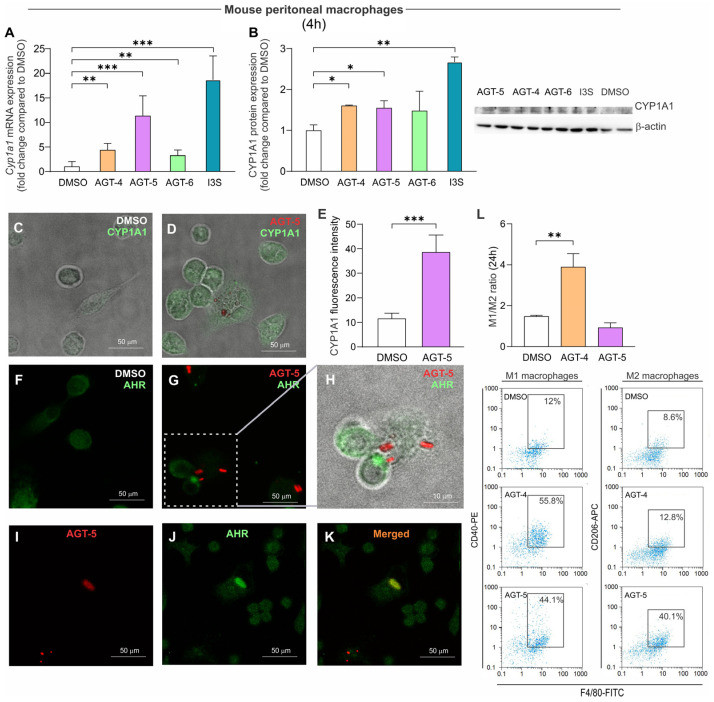
AHR modulation and its impact on macrophage polarization. The effect of FluoAHRL on peritoneal macrophage Cyp1a1 mRNA expression and differentiation. Cyp1a1 mRNA expression (normalized to the expression of β-actin and then normalized to the values obtained in DMSO-treated cells) (**A**) or protein expression (**B**) in mouse peritoneal macrophages 4 h after the exposure to FluoAHRL (1.5 µM), DMSO or I3S (1.5 µM). Representative blots are shown. Merged bright-field and fluorescence confocal microscope images of CYP1A1 (stained green) in DMSO- (**C**) and AGT-5-treated (red) (**D**) peritoneal macrophages after 24 h of culture (orig. magnification 63×). CYP1A1 expression was determined by analyzing fluorescence intensity with Leica LAS AF lite 3.3.0 software (**E**). Fluorescence images of DMSO- (**F**) or AGT-5-treated (**G**) macrophages and merged bright-field and fluorescence confocal microscope image (**H**) of AHR (stained green) and AGT-5 (red) in the peritoneal macrophages after 24 h of culture. Signals from AGT-5 (red) (**I**) and AHR (green) (**J**) signals were merged (orange) (**K**). (**L**) Peritoneal cells were treated with 1.5 µM of FluoAHRL or DMSO, and M1 (F4/80^+^CD40^+^) and M2 (F4/80^+^CD206^+^) macrophage phenotypes were detected by flow cytometry after 24 h of culture. Representative dot plots are shown below the graph. * *p* < 0.05, ** *p* < 0.01, *** *p* < 0.001 was considered as a statistically significant difference between values obtained from FluoAHRL-treated cells and DMSO-treated cells.

**Figure 5 molecules-29-02988-f005:**
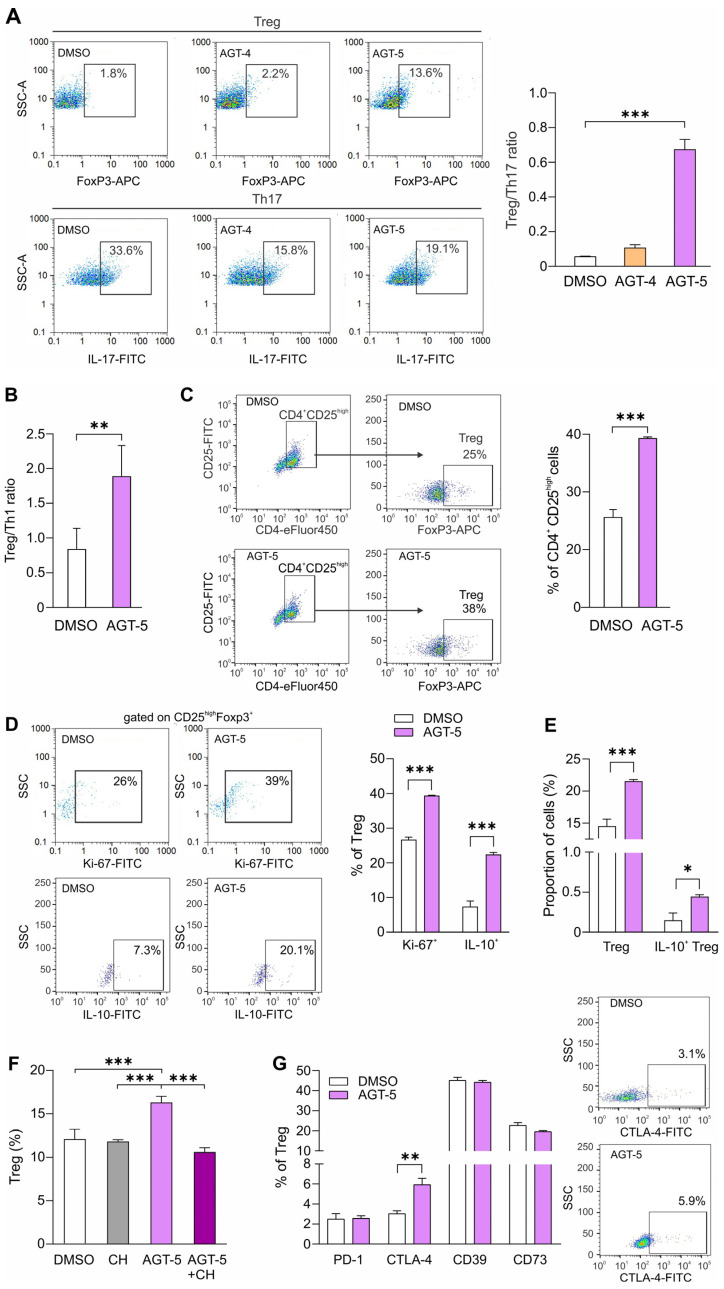
The impact of AGT-4 and AGT-5 on T cell differentiation. Purified CD4^+^ cells were stimulated with anti-CD3 and anti-CD28 antibodies and exposed to FluoAHRL (1.5 µM) or DMSO for 48 h, after which Th17 cells (IL-17^+^) and Treg (FoxP3^+^) were detected on the flow cytometer and presented as Treg/Th17 ratio (**A**). CD4^+^ were exposed to anti-CD3 and anti-CD28 antibodies and treated with AGT-5 (1.5 µM) or DMSO for 48 h, and Th1 (IFN-γ^+^) and Treg (CD25^high^FoxP3^+^) profiles were evaluated and presented as Treg/Th1 ratio (**B**). Naïve CD4^+^CD25^-^ cells were stimulated with AGT-5 for 48 h only in the presence of the anti-CD3 antibody, after which the Treg proportion (**C**), their proliferation (Ki-67^+^) and IL-10 production (**D**) were evaluated (representative dot plots on the right-hand side). Sorted CD4^+^CD25^high^ were treated with AGT-5 for 48 h in the presence of the “complete” stimulation cocktail and the proportions of Treg (CD4^+^CD25^high^FoxP3^+^) and IL-10^+^ Treg were determined by flow cytometry (**E**) (representative dot plots on the right-hand side). CD4^+^CD25^−^ cells were stimulated by anti-CD3 and anti-CD28 antibody and treated with AGT-5 (1.5 µM) in the presence or absence of AHR inhibitor CH-223191 (CH, 1.5 µM) and the proportion of Treg was determined (CD4^+^CD25^high^FoxP3^+^) (**F**). Sorted CD4^+^CD25^high^ were treated with AGT-5 for 48 h in the presence of the “complete” stimulation cocktail and the proportions of Treg expressing PD-1, CTLA-4, CD39 and CD73 were ascertained (**G**). Representative dot plots for CTLA-4^+^ Treg are shown. * *p* < 0.05, ** *p* < 0.01, *** *p* < 0.001 was considered as a statistically significant difference between AGT-5-treated cells and DMSO-treated cells, or between AGT-5+CH-223191-treated cells and AGT-5-treated cells.

**Figure 6 molecules-29-02988-f006:**
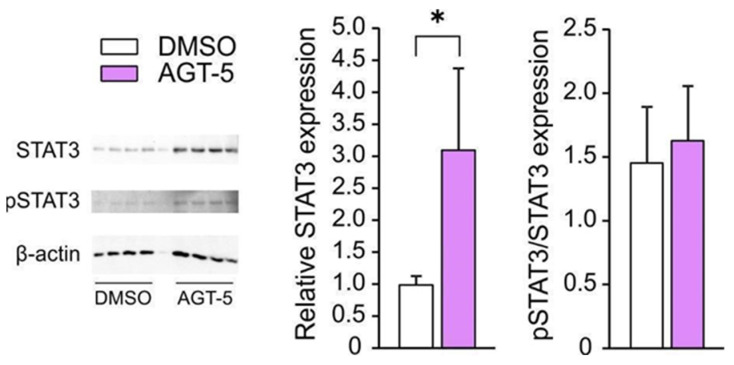
AGT-5 impact on T cell signaling. CD4+ were exposed to anti-CD3 and anti-CD28 antibodies and treated with AGT-5 or DMSO for 24 h. STAT3 protein expression (relative to β-actin) and pSTAT3/STAT3 ratio were determined by western blot. Representative blots are shown (4 samples for DMSO and 4 samples for AGT-5; the well in between contains spillover from adjacent wells). * *p* < 0.05 was considered as a statistically significant difference between AGT-5-treated cells and DMSO-treated cells.

**Figure 7 molecules-29-02988-f007:**
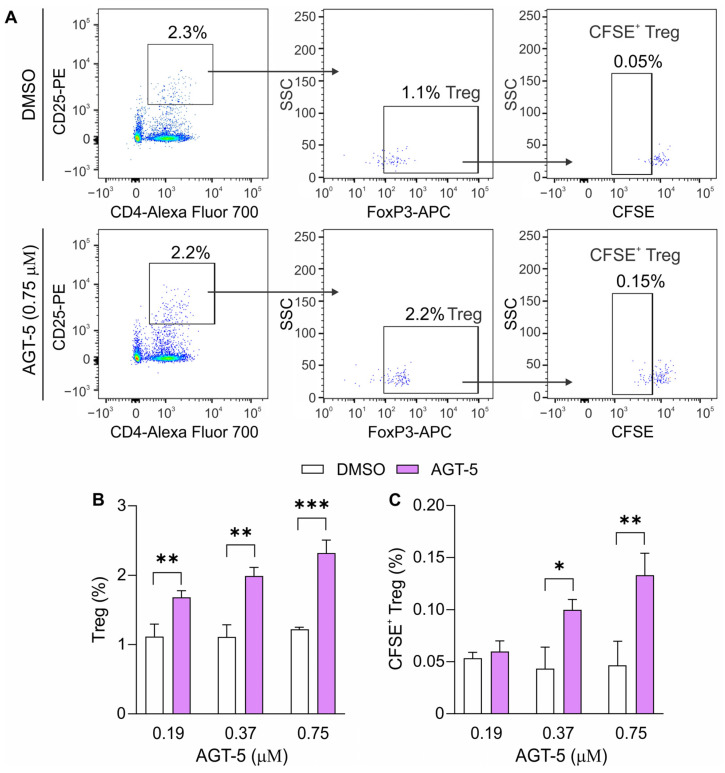
AGT-5 impact on human Treg differentiation and proliferation. Human tonsillar cells were cultured in the presence of increasing concentrations of AGT-5 or DMSO and evaluated for the Treg proportion (**B**) and proliferating (CFSE^+^) Treg (**C**) with flow cytometry after 48 h of cultivation. Representative dot plots (for 0.75 µM dose of AGT-5) are shown (**A**). * *p* < 0.05, ** *p* < 0.01, *** *p* < 0.001 was considered as a statistically significant difference between AGT-5-treated cells and DMSO-treated cells.

**Figure 8 molecules-29-02988-f008:**
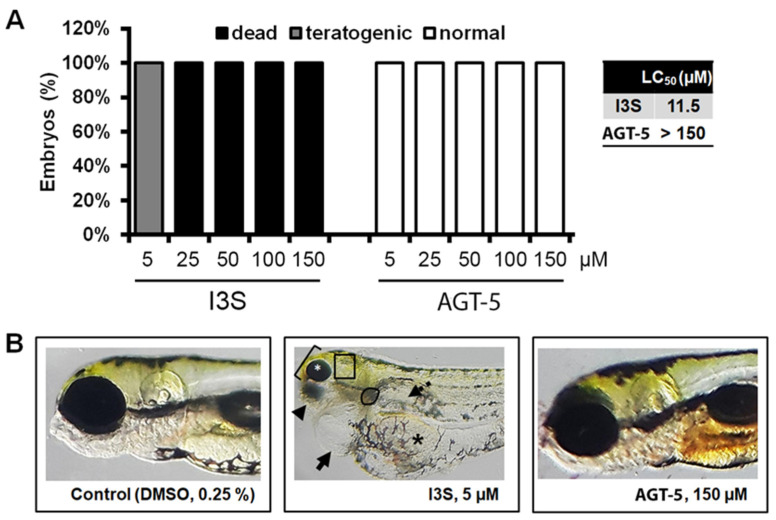
Toxicity assessment of AGT-5 and I3S in zebrafish. The survival/teratogenicity (**A**) and morphology (**B**) of zebrafish embryos exposed to different doses of the selected compounds (5, 25, 50, 100 and 150 µM) at 120 h post fertilization (hpf) are shown. In contrast to AGT-5 treatment, due to I3S exposure, live embryos suffered from severe pericardial edema (black arrow), hepatotoxicity (liver necrosis—outlined area; non-resorbed egg yolk—black asterisk), nephrotoxicity (dashed arrow), as well as malformation of the head (bracket), jaw (arrowhead) and eyes (white asterisk).

**Figure 9 molecules-29-02988-f009:**
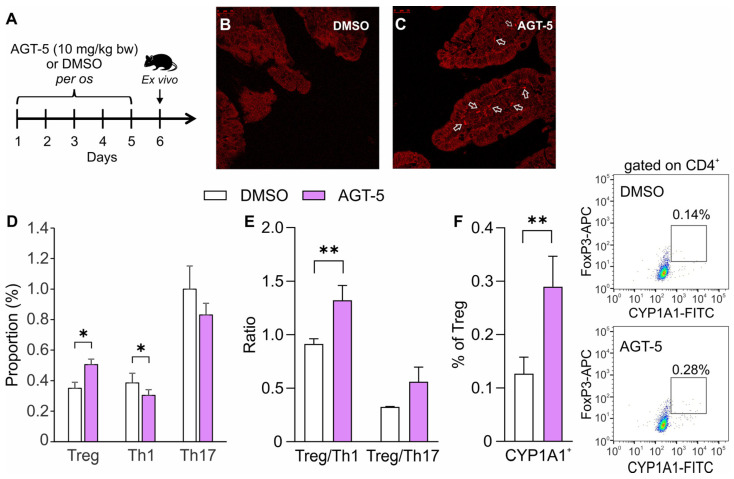
AGT-5 in vivo elicited responses. Administration of AGT-5 to healthy C57BL/6 mice. AGT-5 (10 mg/kg bw) or DMSO were administered orally for five days (**A**). Confocal microscopy images of small intestine sections of DMSO-treated (**B**) and AGT-5-treated animals (**C**) (the distribution of AGT-5 is indicated by white arrows). Th1, Th17 and Treg were determined in the mesenteric lymph nodes after oral AGT-5 administration (**D**) and also presented as Treg/Th1 and Treg/Th17 ratio (**E**). The proportion of CYP1A1^+^ cells within Treg (**F**). The representative dot plots show CYP1A1^+^FoxP3^+^ cells that are already gated on CD4^+^CD25^high^. * *p* < 0.05, ** *p* < 0.01 was considered as a statistically significant difference between AGT-5-treated mice and DMSO-treated mice.

## Data Availability

The data presented in this study are available on request from the corresponding author.

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
