# Peer review of "Development of FluoAHRL: A Novel Synthetic Fluorescent Compound That Activates AHR and Potentiates Anti-Inflammatory T Regulatory Cells"

_molecules, 2024, doi:10.3390/molecules29132988_

Round 1

Reviewer 1 Report

Comments and Suggestions for Authors

Submitted manuscript presents data demonstrating that three novel molecules synthesized by the authors act as AHR agonists, present fluorescence in the near-infrared and display anti-inflammatory activity. The experiments were very well planned and included detailed studies in vitro and ex vivo, with the use of appropriate techniques. The data were properlu anlyzed and are sound. Congratulations to the authors!

Author Response

REVIEWER 1

Submitted manuscript presents data demonstrating that three novel molecules synthesized by the authors act as AHR agonists, present fluorescence in the near-infrared and display anti-inflammatory activity. The experiments were very well planned and included detailed studies in vitro and ex vivo, with the use of appropriate techniques. The data were properly analyzed and are sound. Congratulations to the authors!

AU: We would like to express our sincere appreciation to the reviewer for the thoughtful and encouraging feedback on our work, particularly on this manuscript.

Reviewer 2 Report

Comments and Suggestions for Authors

AHR is a transcriptional factor involving differentiation of Treg/T17 cells and affecting maturation and polarization of antigen presenting cells. The study aimed to develop novel fluorescent AHR ligands with anti-inflammatory function. Such molecules would be useful for molecular tracking of ligand/AHR interaction and, may contribute to find a novel mechanism of AHR-mediated immune modulation. The manuscript is well written. However, it would be better to add several texts in introduction to highlight the novelty of this study clearly. In addition, to improve the quality of data, the authors are suggested to add proper controls in several experiments.

(1) Fig. 4 and Fig. 6: AGT-4, AGT-5, and AGT-6 are fluorescence. The fluorescence does not affect results of FACS analysis using FITC, PE and APC channels?

(2) Fig. 4 and Fig. 6: It would be important to see whether fluorescence of AGT-4, AGT-5, or AGT-6 is detected in CD40 high or CD206 high F4/80 positive cells, or Foxp3 positive cells, i.e. in ligand-affected cells, under a proper compensation condition. If the cells take the AHR ligand up, the fluorescence derived from the ligand must be detected?

(3) Please include data of isotype antibody controls in all FACS analysis.

(4) Fig.4G: The authors described that AHR is accumulated in the proximity to AGT-5 in cells treated with this compound (Figure 4F-H). Can this proximity of AHR to AGT-5 be observed in the majority of treated cells? Only one image focusing a few cells would not support the result. Please also include explanation why AGT-5 does not “co-localize” with AHR. The AHR ligand does not enter to nucleus?

(5) Fig.4B: Quality of western blotting image is low. Please show blotting data with clear signals.

(6) Figs.4I, 5 and 6: Please include data of a positive control, well established AHR ligand such as FICZ, to assess the activity of AGT-4 and AGT-5 in induction of M1/M2 or Treg/Th17 cells, properly.

Author Response

REVIEWER 2

AHR is a transcriptional factor involving differentiation of Treg/T17 cells and affecting maturation and polarization of antigen presenting cells. The study aimed to develop novel fluorescent AHR ligands with anti-inflammatory function. Such molecules would be useful for molecular tracking of ligand/AHR interaction and, may contribute to find a novel mechanism of AHR-mediated immune modulation. The manuscript is well written. However, it would be better to add several texts in introduction to highlight the novelty of this study clearly. In addition, to improve the quality of data, the authors are suggested to add proper controls in several experiments.

AU: We sincerely appreciate the reviewer's positive feedback and kind words. We have thoughtfully integrated the reviewer's suggestions and comments, leading to an enhanced version of the manuscript.

It would be better to add several texts in introduction to highlight the novelty of this study clearly. 

AU: We acknowledge the reviewer’s comment. We have updated the manuscript to reflect the reviewer’s feedback and included a section discussing the novelty of this study (see lines 104-110).

  • 4 and Fig. 6: AGT-4, AGT-5, and AGT-6 are fluorescence. The fluorescence does not affect results of FACS analysis using FITC, PE and APC channels?

AU: We would like to thank the reviewer for the insightful comment. In each in vitro experiment, we used an internal control, i.e., unstained cells treated with the specific FluoAHRL. According to flow cytometry analysis, all three FluoAHRLs did not emit signals in the FITC, PE, and APC channels (see attached Figures 1 A-C and 2 A-C). However, AGT-6 showed considerable emission at 710 and 780 nm (see attached Figures 1D, E, and 2D, E). It is important to note that AGT-6 emission at these wavelengths did not interfere with other assays presented here, such as the luciferase assay (emission spectra between 520 and 570 nm) and gene expression analysis by RT-qPCR. Nevertheless, AGT-6 did impact AHR activity in different cell types (as shown in Figures 3, 4, and S19). Albeit, due to cellular toxicity in peritoneal macrophages (see Figure S20 in the revised manuscript), AGT-6 was not used to evaluate its immunomodulatory properties.

Altogether, we consider that the fluorescence properties of the described ligands (AGT-4 and AGT-5) did not interfere with specific antibody stainings or the conclusions drawn from these experiments, at the conditions tested. As mentioned to the reviewer, the FACS control plots for all channels are now included in the revised manuscript as Figures S21 and S22, with a brief explanation added to the results section (lines 288-290 and 334-335).

  • 4 and Fig. 6: It would be important to see whether fluorescence of AGT-4, AGT-5, or AGT-6 is detected in CD40 high or CD206 high F4/80 positive cells, or Foxp3 positive cells, i.e. in ligand-affected cells, under a proper compensation condition. If the cells take the AHR ligand up, the fluorescence derived from the ligand must be detected?

AU: We appreciate the reviewer's comment. To address this important question, similar to what was described in point 1 (see above), we have updated the manuscript to include new figures (Figure S21 and S22) showing the controls used in these experiments, specifically the FACS plots from unstained cells treated with the specific FluoAHRLs. We have also updated the text accordingly (lines 288-290 and 334-335). In brief, we could not detect the fluorescence of AGT-4 and AGT-5 by flow cytometry analysis in the channels used to analyze the immunomodulatory properties of these compounds in our in vitro experiments (shown in attached Figures 1 and 2). Remarkably, AGT-5 fluorescence can be visualized in tissues ex vivo from mice treated with this ligand, for example, in the small intestine (Figure 9C). However, consistent with the in vitro data, flow cytometry analysis did not detect fluorescent cells in ex vivo samples (e.g. mesenteric lymph nodes, lamina propria, and spleen) from mice after oral application of AGT-5 (data not shown).

  • Please include data of isotype antibody controls in all FACS analysis.

AU: Following the reviewer’s suggestion, in the revised version of the manuscript we provide a figure with the isotype antibody controls (Figure S23). Accordingly, we updated the manuscript text (lines 335-338; 830-831).

  • 4G: The authors described that AHR is accumulated in the proximity to AGT-5 in cells treated with this compound (Figure 4F-H). Can this proximity of AHR to AGT-5 be observed in the majority of treated cells? Only one image focusing a few cells would not support the result. Please also include explanation why AGT-5 does not “co-localize” with AHR. The AHR ligand does not enter to nucleus?

AU: We would like to thank the reviewer for these comments. In the revised version of the manuscript we now provide additional images to support the results of the accumulation of AHR close to AGT-5. As can be seen in Figure 3A-C (attached), and in Figures 4F-K of the main manuscript, a marked accumulation of AHR (green) signal is detected close to AGT-5 (red). Furthermore, this accumulation can be completely colocalized with the signal detected for AGT-5, in some of the cells containing AGT-5, as depicted in Figures 4I-K of the main manuscript (corresponding to Figures 3A-C - attached). Remarkably, and reassuring that indeed this is not related to bleed-through of the AGT-5 signal to the AHR channel, we can observe bright spots of AGT-5 signal in the red channel (Figures 4I, K bottom left corner) and no respective signal on the green channel (Figures 4J-K). We have now updated the manuscript to include these images and revised the text accordingly (lines 273-276; 323-324). Concerning the localization of AHR and AGT-5, unfortunately, we cannot confirm nuclear localization as we did not stain for the nucleus (e.g. DAPI) due to limitations of our confocal microscope equipment. Nevertheless, when re-analyzing the images obtained to try to still address this point, we noticed different patterns of spatial co-localization/signal accumulation (see attached Figures 3D-G ). Altogether, and as further experiments with additional controls, and a systematic approach to be able to quantify nuclear versus cytoplasmic ratios under different conditions (exposure time, different concentration of ligands) would be needed to fully address this point, we refrain from making conclusions about the nuclear localization of the ligand.

  • 4B: Quality of western blotting image is low. Please show blotting data with clear signals.

AU: We agree with the reviewer’s comment. We have made efforts to improve the quality of the Western Blot image and have replaced it in the revised version of the manuscript (see Figure 4B). However, the quality of the antibody limited our ability to make significant improvements. If the reviewer and the editor prefer, we are open to removing this image from the manuscript, as we have demonstrated increased CYP1A1 expression in the presence of the FluoAHRLs through other methods (e.g. RT-qPCR, IF).

(6) Figs.4I, 5 and 6: Please include data of a positive control, well established AHR ligand such as FICZ, to assess the activity of AGT-4 and AGT-5 in induction of M1/M2 or Treg/Th17 cells, properly.

AU: We thank the reviewer’s comment. We apologize for any potential misunderstanding. In our experiments depicted in Figures 4 and 5, we used the uremic toxin 3-indoxyl sulfate (I3S)1 as a positive control or the AHR antagonist CH2231912 as an additional control. We have updated the text to clarify this information (see lines 268-270; 343). Notably, similar to FICZ, I3S is a well-established AHR ligand3. Schroeder et al demonstrated that I3S can bind to and activate the AHR in human and mouse cells1. Due to its AHR-elicited immunomodulatory effects, i3S has been widely used in studies related to AHR and inflammation, experimental autoimmune encephalomyelitis, asthma, and Th cell differentiation, among others3-6. It is widely known that AHR responses are highly context-dependent, with exposure conditions, ligands, cells, tissues, and organisms (e.g., mouse versus human differences) all playing a role3,7,8. For example, differences in Th17 and Treg differentiation have been observed upon AHR stimulation by FICZ or TCDD3,7,9,10. Our results show that similar to AGT-5, the chosen AHR agonist control (I3S) also supports Treg differentiation (Figure S24). Furthermore, CH223191 (AHR antagonist)2 could reverse this phenotype in AGT-5 exposed cells (Figure 5F; mentioned in lines 354-356). Altogether, even though different AHR agonists may elicit different responses, it is beyond the scope of the current study to evaluate this systematically, and we consider that I3S lists as a relevant positive control to activate the AHR and to support Treg differentiation (Figure S24).

Attached Figure 1. Peritoneal cells under FluoAHRL exposure. Exposure of peritoneal cells to the FluoAHRL (1.5 µM) for 48 h and detection of fluorescent signal in all channels that were used for detection of specific staining with antibodies. (A) FITC channel (B525 nm). (B) PE channel (B585 nm). (C) APC channel (R660 nm). (D) APC-eF780 channel (R780 nm). (E) PercP-eF710 channel (B710 nm). (F) eFluor450 channel (V450 nm).

Attached Figure 2. Figure S5. Mesenteric lymph node cells under FluoAHRL exposure. Exposure of mesenteric lymph node cells to the FluoAHRL (1.5 µM) for 48 h and detection of fluorescent signal in all channels that were used for detection of specific staining with antibodies. (A) FITC channel (B525 nm). (B) PE channel (B585 nm). (C) APC channel (R660 nm). (D) APC-eF780 channel (R780 nm). (E) PercP-eF710 channel (B710 nm). (F) eFluor450 channel (V450 nm).

Attached Figure 3. Peritoneal cells (macrophages) were cultured for 24 h with AGT-5, what was followed by the AHR staining. A) AGT-5 channel; B) AHR channel; C) Merged image from two channels showing yellow-orange signal from co-localization; D-F Representative images from several wells showing the existence of co-localization of AGT-5 and AHR signal; G) Zoomed image of cells showing different patterns of spatial co-localization/signal accumulation.

References

1          Schroeder, J. C. et al. The uremic toxin 3-indoxyl sulfate is a potent endogenous agonist for the human aryl hydrocarbon receptor. Biochemistry 49, 393-400, doi:10.1021/bi901786x (2010).

2          Zhao, B., Degroot, D. E., Hayashi, A., He, G. & Denison, M. S. CH223191 is a ligand-selective antagonist of the Ah (Dioxin) receptor. Toxicol Sci 117, 393-403, doi:10.1093/toxsci/kfq217 (2010).

3          Rothhammer, V. & Quintana, F. J. The aryl hydrocarbon receptor: an environmental sensor integrating immune responses in health and disease. Nat Rev Immunol 19, 184-197, doi:10.1038/s41577-019-0125-8 (2019).

4          Rothhammer, V. et al. Microglial control of astrocytes in response to microbial metabolites. Nature 557, 724-728, doi:10.1038/s41586-018-0119-x (2018).

5          Rothhammer, V. et al. Type I interferons and microbial metabolites of tryptophan modulate astrocyte activity and central nervous system inflammation via the aryl hydrocarbon receptor. Nat Med 22, 586-597, doi:10.1038/nm.4106 (2016).

6          Hwang, Y. J., Yun, M. O., Jeong, K. T. & Park, J. H. Uremic toxin indoxyl 3-sulfate regulates the differentiation of Th2 but not of Th1 cells to lessen allergic asthma. Toxicol Lett 225, 130-138, doi:10.1016/j.toxlet.2013.11.027 (2014).

7          Stockinger, B., Di Meglio, P., Gialitakis, M. & Duarte, J. H. The aryl hydrocarbon receptor: multitasking in the immune system. Annu Rev Immunol 32, 403-432, doi:10.1146/annurev-immunol-032713-120245 (2014).

8          Di Meglio, P. et al. Activation of the aryl hydrocarbon receptor dampens the severity of inflammatory skin conditions. Immunity 40, 989-1001, doi:10.1016/j.immuni.2014.04.019 (2014).

9          Quintana, F. J. et al. Control of T(reg) and T(H)17 cell differentiation by the aryl hydrocarbon receptor. Nature 453, 65-71, doi:10.1038/nature06880 (2008).

10        Veldhoen, M. et al. The aryl hydrocarbon receptor links TH17-cell-mediated autoimmunity to environmental toxins. Nature 453, 106-109, doi:10.1038/nature06881 (2008).

Reviewer 3 Report

Comments and Suggestions for Authors

This study investigates the effects of three synthetic Aryl Hydrocarbon Receptor (AHR) ligands (FluoAHRL: AGT-4, AGT-5, and AGT-6) on the modulation of pro- and anti-inflammatory responses. All synthesized compounds were designed to emit fluorescence in the near-infrared spectrum. Their AHR agonist activity was first predicted using in silico docking studies and then confirmed using AHR luciferase reporter cell lines. FluoAHRLs were tested in vitro with mouse peritoneal macrophages and T lymphocytes to assess their immunomodulatory properties. Among them, AGT-5 predominantly exhibited anti-inflammatory effects.

The study was well designed, and the data support the hypothesis. However, the imaging quality in Figure 9B, Figure 9C, and the western blot in Figure 4B is poor. The authors should replace these images with higher-quality versions.

Comments on the Quality of English Language

Minor editing of English language required

Author Response

REVIEWER 3

This study investigates the effects of three synthetic Aryl Hydrocarbon Receptor (AHR) ligands (FluoAHRL: AGT-4, AGT-5, and AGT-6) on the modulation of pro- and anti-inflammatory responses. All synthesized compounds were designed to emit fluorescence in the near-infrared spectrum. Their AHR agonist activity was first predicted using in silico docking studies and then confirmed using AHR luciferase reporter cell lines. FluoAHRLs were tested in vitro with mouse peritoneal macrophages and T lymphocytes to assess their immunomodulatory properties. Among them, AGT-5 predominantly exhibited anti-inflammatory effects.

The study was well-designed, and the data support the hypothesis. However, the imaging quality in Figure 9B, Figure 9C, and the western blot in Figure 4B is poor. The authors should replace these images with higher-quality versions.

AU: We would like to thank the reviewer for the careful reading of our manuscript and the kind words and positive feedback. We agree with the reviewer’s comment on the quality of the images and have made efforts to enhance the quality of the Western Blot image, which has been replaced in the revised manuscript (see Figure 4B). However, the quality of the antibody limited the extent of our improvements. If the reviewer and editor prefer, we are willing to remove this image, as we have demonstrated increased CYP1A1 expression in the presence of FluoAHRLs through other methods (e.g. RT-qPCR, IF). Regarding Figures 9B and 9C, they have been replaced with higher-quality images in the revised manuscript.

Round 2

Reviewer 2 Report

Comments and Suggestions for Authors

The authors replied to the comments by this reviewer well and improved the quality of manuscript.